

# Jet substructure observables for jet quenching in quark gluon plasma: A machine learning driven analysis

Miguel Crispim Romão[1,2⋆], José Guilherme Milhano[1,3] and Marco van Leeuwen[4]

**1** Laboratório de Instrumentação e Física Experimental de Partículas (LIP),
Av. Professor Gama Pinto 2, 1649-003 Lisboa, Portugal
**2** Department of Physics and Astronomy, University of Southampton,
SO17 1BJ Southampton, United Kingdom
**3** Departamento de Física, Instituto Superior Técnico, Universidade de Lisboa,
Av. Rovisco Pais 1, 1049-001 Lisboa, Portugal
**4** Nikhef, National Institute for Subatomic Physics, P.O. Box 41882, 1009 DB Amsterdam and
Utrecht University, P.O. Box 80000, 3508 TA Utrecht, The Netherlands

⋆ mcromao@lip.pt

## Abstract

We present a survey of a comprehensive set of jet substructure observables commonly used to study the modifications of jets resulting from interactions with the Quark Gluon Plasma in Heavy Ion Collisions. The JEWEL event generator is used to produce simulated samples of quenched and unquenched jets. Three distinct analyses using Machine Learning techniques on the jet substructure observables have been performed to identify both linear and non-linear relations between the observables, and to distinguish the Quenched and Unquenched jet samples. We find that most of the observables are highly correlated, and that their information content can be captured by a small set of observables. We also find that the correlations between observables are resilient to quenching effects and that specific pairs of observables exhaust the full sensitivity to quenching effects. The code, the datasets, and instructions on how to reproduce this work are also provided.



# 1  Introduction

In ultra-relativistic collisions of protons and nuclei at the Large Hadron Collider, high-energy quarks and gluons are produced in hard QCD scatterings, as well as in the decays of electroweak bosons. Their experimental signature are jets of hadrons, which are reconstructed using jet algorithms [1]. Recently, there has been growing interest in the study of jet substructure, both as a tool to test the Standard Model and to search for physics beyond it [2–9], and as a way to map in detail the QCD branching process [10–13].

In the context of heavy-ion collisions, the interaction between partons in the QCD branching sequence and the Quark Gluon Plasma (QGP) results in modifications of what is ultimately reconstructed as a jet (see [14–17] for reviews). Jet shape observables are of particular interest [18] because they characterise jet substructure on a jet-by-jet basis and therefore have the potential to establish a correspondence between specific changes of reconstructed jets and specific features of the interaction with the QGP (modifications of the branching sequence [19–57], transfer of energy-momentum from partons to QGP [58, 59], and QGP response [60–70]).

Establishing a direct connection between jet shape observables and these mechanisms has so far proven elusive. In this paper, we present a survey of a large set of jet observables, including jet shape observables, that have been suggested in the literature using events generated with the JEWEL event generator with and without jet quenching effects.

In recent years high energy physics (HEP) has seen a resurgence of interest in machine learning and artificial intelligence (AI/ML) techniques, fuelled by the advent of modern deep learning (DL), that are now applied to a wide range of tasks in HEP [71]. In the context of jets in heavy-ion collisions, machine learning techniques have been used for the identification of quenching effects and discrimination between jets produced in medium from those produced in vacuum [72–75] and to explore geometrical aspects relevant for jet quenching, i.e. jet tomography [72,76,77]. Further applications in the context of heavy ion collisions are reviewed in [78].

Most previous studies have privileged the use of low-level data (e.g. jet images) which hold the potential to contain the whole information about jet substructure [7] while a machine learning driven analysis of high-level jet substructure variables, several of which are directly motivated by theoretical arguments and are commonly reported by the experiments, has still been missing in the literature. This work aims to fill that gap.

This paper is organised as follows. In Section 2 we introduce the observables and the operating definitions that we will use in our study. In Section 3 we discuss the simulation details, namely medium settings and analysis cuts. In Section 4 we will present our first analysis, which focuses on linear correlations between observables and Principal Components analysis, in order to identify the main groups of independent observables. In Section 5 we go a step further to present a similar analysis but using a Deep Auto-Encoder, which can learn non-linear relations between observables. In Section 6 we produce an analysis using the discrimination between Unquenched and Quenched samples provided by the different observables to further understand which can be sensitive to medium effects. In Section 7 we present an exploratory study of the impact of QGP response in the Quenched sample. Finally, in Section 8 we conclude. All the data and code used to produce our analyses is publicly provided, see Appendix A for instructions on how to reproduce this work.

## 2 Observables

In this section we introduce the observables explored in this work. We will introduce the notation and the definition of each observable noting, in particular, how they may differ from elsewhere in the literature. We consider only observables which return a single value per jet. That is to say, we do not consider distributions such as the jet profile but rather moments of such a distribution. We have grouped the considered observables as follows.

### 2.1 Jet momenta and constituent multiplicity

The first set of observables includes the jet total 4-momentum and the total number of constituents of a jet (the number of particles reconstructed as part of a jet) $n_{\text{const}}$.

We write the jet 4-momentum in terms of an azimuthal angle $\phi_{\text{jet}}$ and transverse momentum $p_{\text{T,jet}}$ for the transverse momentum components, the rapidity $y_{\text{jet}} = -\ln\frac{1}{2}\left(\frac{E_{\text{jet}}+p_{\text{jet,z}}}{E_{\text{jet}}-p_{\text{jet,z}}}\right)$ as the longitudinal observable, and the mass $m_{\text{jet}} = \sqrt{p \cdot p}$. That is

$$
\begin{aligned}
p_{\text{jet},\mu} &= (E_{\text{jet}}, p_{\text{jet,x}}, p_{\text{jet,y}}, p_{\text{jet,z}}) \\
&= (\sqrt{m_{\text{jet}}^2 + p_{\text{T,jet}}^2}\cosh y_{\text{jet}}, p_{\text{T,jet}}\cos\phi_{\text{jet}}, p_{\text{T,jet}}\sin\phi_{\text{jet}}, \sqrt{m_{\text{jet}}^2 + p_{\text{T,jet}}^2}\sinh y_{\text{jet}}).
\end{aligned}
\tag{1}
$$

The jet 4-momentum is calculated from the 4-momenta of constituents using the *E-scheme* recombination, which is the standard (4-)vector sum.

### 2.2 Angularities

The second set of observables are those derived from the generalised angularities [10], i.e. moments of the distribution of jet constituents around the jet axis

$$
\lambda_\beta^\kappa = \sum_{i \in jet} z_i^\kappa R_{i,jet}^\beta.
\tag{2}
$$

Here, $z_i$ is the fraction of the jet transverse momentum carried by the constituent $i$, $z_i = p_{T,i}/p_{T,jet}$ and $R_{i,jet}$ is the angular distance of the constituent to the jet axis.

The angular exponent $\beta \geq 0$ accounts for weighting with distance to the axis, while $\kappa \geq 0$ is an energy weighting factor. The angular distance between any two 4-momenta, $i$ and $j$, is given by

$$R_{i,j} = \sqrt{(y_i - y_j)^2 + (\phi_i - \phi_j)^2}\,. \tag{3}$$

The generalised angularities (with $\beta > 0$) describe the angular distribution of the momentum flow in the jet, and are therefore associated with the transverse properties of the jet fragmentation. Note that in the definition presented here and used throughout this study, the angular distances are not divided by the jet radius.

These observables are only Infrared and Collinear (IRC) safe for $\kappa = 1$, since then the sum over the momentum fractions $\Sigma z_i = 1$. For $\kappa \neq 1$, the observable becomes explicitly dependent on multiplicity since $\Sigma z_i^{\kappa_i \neq 1} \neq 1$. For the $\kappa = 0$ cases we will explicitly divide by $n_{\text{const}}$ to obtain an average over the constituents.

Since the main interest of the angularities is to capture the transverse characteristics of the jet, we will only consider $\beta \neq 0$, with the parameters $\kappa = 0$ and 1, and $\beta = 1$ and 2.

In addition, we will use the momentum dispersion, $p_{T,D}$ [79], which measures the second moment of the constituent $p_T$ distribution in the jet and is connected to how hard or soft the jet fragmentation is

$$p_{T,D} = \frac{\sqrt{\sum_{i \in jet} p_{T,i}^2}}{p_{T,jet}} = \sqrt{\lambda_0^2}\,. \tag{4}$$

Since this quantity is not IRC safe due to the fact that $\kappa \neq 1$, we will also consider the mean over consitutuents of its square, i.e.

$$\bar{z}^2 = \frac{1}{n_{\text{const}}} \lambda_0^2 = \frac{1}{n_{\text{const}}} \sum_{i \in jet} z_i^2\,. \tag{5}$$

## 2.3 *N*-subjettiness

Another set of observables that captures transverse properties of a jet is the $N$-Subjettiness [80]. These observables measure how similar a given jet is to an object composed of $N$ subjets. They read, for a given number $N$ of candidate subjets,

$$\tau_N = \frac{\sum_{i \in jet} p_T^i \min(R_{1,i}, \ldots, R_{N,i})}{R_0\, p_{T,jet}}\,, \tag{6}$$

where $R_0$ is the jet radius used in the jet clustering algorithm and $R_{j,i}$ is the distance between constituent $i$ and subjet $j$.

These observables characterise the jet in terms of the spread around $N$ subjets; low values of $\tau_N$ indicate an $N$-subjet like particle distribution. However, the values of $\tau_N$ also depend on multiplicity, and a better measurement of the number of subjets is then given by ratios of $\tau_N$ for different $N$, i.e.

$$\tau_{N,N-1} = \frac{\tau_N}{\tau_{N-1}}\,, \tag{7}$$

with the smaller the value, the more $N$-subjet like the constituent distribution is.

It is worth noting that $\tau_1$ is equal to the (1,1)-angularity $\lambda_1^1$, which we refer to as $rz$ in this paper. In this analysis, we will therefore focus on $N = 2, 3$ and the ratios $\tau_{2,1}$, $\tau_{3,2}$.

## 2.4 Jet charges

Another jet observable that has been measured [81, 82] is the Jet Charge [83], given by

$$Q^\kappa = \sum_{i \in jet} z_i^\kappa Q_i\,, \tag{8}$$

where the sum is over all of the charges $Q_i$ and transverse momentum fractions $z_i$ of all the jet particles. Because charge is conserved in both parton splittings and hadronization, jet charge is a proxy measure of the charge of the initial parton, and can therefore serve to distinguish quark and gluon jets. The jet charge distribution for gluon jets peaks at zero, while it peaks at finite, but opposite, values for quarks and anti-quarks. To avoid the quark/anti-quark ambiguity, we will be using the absolute value of the jet charge. We will focus on values of $\kappa$ that have been studied by experiments: $\kappa = 0.3,\ 0.5,\ 0.7,\ 1.0$.

## 2.5 Grooming techniques

The observables presented so far are obtained from the list of particles which are clustered into a jet. Here, we will introduce observables obtained after those jets are subjected to a grooming procedure.

### 2.5.1 SoftDrop

SoftDrop [84] is a widely used grooming technique that takes the constituents of a jet (usually reclustered with the C/A algorithm), and recursively declusters the jet branching history discarding the softest branch (subjet) until the transverse momenta of the current pair fulfill the condition

$$\frac{\min[p_{T,i}, p_{T,j}]}{p_{T,i} + p_{T,j}} > z_{cut} \left(\frac{R_{i,j}}{R_0}\right)^{\beta}, \tag{9}$$

where $R_{i,j}$ is the angular distance between the two subjets, $z_{cut}$ and $\beta$ are parameters which select how strict the grooming procedure is, and $R_0$ is the jet radius used for the initial clustering.

For $\beta = 0$, the only setting we will consider here, the grooming procedure corresponds to the modified Mass Drop Tagger (mMDT) [85].

At the splitting that satisfies the SoftDrop condition, one can compute the fraction of the transverse momentum contained within the softer branch

$$z_g = \frac{p_{T,2}}{p_{T,1} + p_{T,2}}, \tag{10}$$

where $p_{T,2}$ ($p_{T,1}$) is the momentum of the softer (harder) branch. In addition, one can compute the distance $R_g$ between the branches at the first declustering step that fulfills the SoftDrop condition and how many splittings satisfy the SoftDrop condition in a recursive declustering, $n_{SD}$ [86].

### 2.5.2 Dynamical grooming

Dynamical grooming is a grooming technique that selects the first C/A reclustering sequence branch that satisfies the condition [13]

$$\kappa^{(a)} = \frac{1}{p_{T,jet}} \max_{i \in \text{C/A seq}} \left[ z_i(1-z_i) p_{T,i} \left(\frac{R_{i,j}}{R_0}\right)^a \right], \tag{11}$$

where $z_i$ the momentum sharing fraction, $p_{T,i}$ the energy of the parent, $R_{i,j}$ the $y - \phi$ distance between the subjets in the splitting and $a$ a free parameter. Depending on the value of $a$, the dynamical grooming quantity $\kappa^{(a)}$ captures different characteristic scales of the C/A clustering sequence:

Table 1: Overview of the set of observables that is considered in the analyses. The subscript $SD$ indicates that the observable was computed from the SoftDrop groomed jet.

| Observable | Type |
|---|---|
| $y_{SD}$ $\phi_{SD}$ $\Delta p_{T,SD} = p_{T,jet} - p_{T,jet_{SD}}$ $m_{SD}$ $n_{\text{const,SD}}$ | Jet Momenta and Constituent Multiplicity |
| $\bar{r}_{SD} = \frac{1}{n_{\text{const,SD}}}\lambda^0_{1,SD}$ $\bar{r}^2_{SD} = \frac{1}{n_{\text{const,SD}}}\lambda^0_{2,SD}$ $rz_{SD} = \lambda^1_{1,SD}$ $r^2 z_{SD} = \lambda^1_{2,SD}$ $\bar{z}^2_{SD} = \frac{1}{n_{\text{const,SD}}}\lambda^2_{0,SD}$ $p_T D_{SD} = \sqrt{\sum_{i\in jet_{SD}} p^2_{T,i}}/p_{T,jet,SD}$ | Angularities |
| $\tau_{2,SD}$, $\tau_{3,SD}$ $\tau_{1,2,SD}$, $\tau_{2,3,SD}$ | $N$-subjettiness |
| $|Q^{0.3}_{SD}|$, $|Q^{0.5}_{SD}|$, $|Q^{0.7}_{SD}|$, $|Q^{1.0}_{SD}|$, | Jet-Charges |
| $R_g$, $z_g$, $n_{SD}$ | SoftDrop Grooming Intrinsic |
| $R_{g,A}$, $z_{g,A}$, $\kappa_A$ with $A \in \{TD, ktD, zD\}$ | Dynamical Grooming Intrinsic |

- TimeDrop (TD): $a = 2$ selects the splitting with the shortest formation time $t_f^{-1} \sim \kappa^{(2)} p_T$.

- $k_T$-Drop (ktD): $a = 1$ tags the splitting with the largest relative transverse momenta $k_T \sim \kappa^{(1)} p_T$.

- $z$-Drop (zD): $a = 0$ corresponds to the splitting with the most symmetric momentum sharing (collinear sensitive, so $a = 0.1$ is used instead).

The selected splitting in the dynamical grooming is characterised by the value of $\kappa^{(a)}$, the momentum fraction $z_g$ and the subjet distance $R_g$. In this work, we will consider the three possible values of $a$ above, and keep the values of $\kappa^{(a)}$, $R_g$, and $z_g$ for each case.

## 2.6 Summary of observables

The final selection of observables to be studied in this work is presented in Table 1, where all the observables are calculated using SoftDrop groomed jets, as indicated by the subscript $SD$. Furthermore, we note that the mass of the groomed jet is obtained directly from the 4-momentum of the groomed jet ($m_{SD} = \sqrt{p_{jet_{SD}} \cdot p_{jet_{SD}}}$), which differs from the Groomed Jet Mass computed from the energies and opening angle of the two sub-jets identified by the SoftDrop procedure as $M_g = 2E_1 E_2 (1 - \cos\theta_{12})$.

Complementary analyses were conducted where all observables not specific to SoftDrop were computed on ungroomed jets. These analyses, which are not shown[1] yield very similar results to the ones discussed here.

---

[1]Analyses using ungroomed jets can be reproduced using the code provided (see Appendix A for details).

# 3 Data simulation details

Samples of simulated jets were generated using JEWEL+PYTHIA (version 2.2.0) [87] for both Unquenched and Quenched cases. The Unquenched sample is generated with no QGP present, using the `jewel-vac` executable, corresponding to the JEWEL description of proton-proton collisions. The Quenched sample, generated with the `jewel-simple` executable, includes QGP effects as modelled by JEWEL. For each case 320 000 events were produced with $\sqrt{s} = 5020$ GeV, $p_T \in [40, 250]$ GeV, $y < 2.5$. For the Quenched case, the medium settings were set to $\tau_i = 0.4$ fm/$c$, $T_i = 440$ MeV, $T_c = 170$ MeV, and centrality $0 - 10\%$. Medium response was not considered for the core analyses presented in this work, but is discussed in the last section. We generated weighted events and the event weights are used consistently in all analyses to retain the desired statistical description of the kinematics.

The simulation was carried out using a docker image containing all the required dependencies. The produced events are stored in `HepMC` [88] format, and processed using FastJet [89], with FastJet Contrib packages `SoftDrop`, `Nsubjettiness`, `AxesDefinition`, `Measure-Definition`. Dynamical grooming was implemented using the code provided by the original authors [13].

The observables were computed on a per-jet level and stored in `ROOT` files using the `TTree` format to be analysed downstream. The docker image with the code used in this analysis is available in an online repository. See Appendix A for instructions on how to reproduce the data samples and the analysis data presented in this work.

For the analyses we select jets with $p_T > 80$ GeV/$c$, and split the data into three equally sized data sets: training, validation, and test set. The training set was used to produce exploratory analysis and train the Machine Learning models. The validation set was used to validate the procedure and for model selection, when applicable. The test set was used solely to produce the final plots and analyses presented in this work. Since each stage relies heavily on the capacity of the data set to provide a strong statistical representation of the underlying processes, the methodology herein is only as good as its weakest link. To ensure similar statistical strength in each of the steps, samples of equal size are used.

For illustration purposes, and to familiarize the reader with expected behaviours, we show in Fig. 1 the distributions of some of the observables intrinsic to dynamical grooming. We see that $\kappa$ observables exhibit pronounced discriminating power between Unquenched and Quenched samples, mainly for the $p_T$-Drop ($a = 1$) and TimeDrop ($a = 2$) prescriptions. We also observe that the opening angle of the splitting that is selected by the dynamical grooming algorithm differs strongly between the different dynamical grooming prescriptions, with $z$-Drop grooming selecting the smallest angles.

In Fig. 2 we show the distributions of a selection of observables, which follow the expected shapes and also show the impact of jet quenching on for example the jet mass, the groomed number of constituents, and the girth, $rz$.

# 4 Linear correlations and principal component analysis

Our first analysis explores the linear correlations among observables within each — Unquenched and Quenched — sample, and how these are affected by jet quenching in the QGP as modelled in JEWEL+PYTHIA. Studying linear correlations among observables allows us to identify which of the observables share the same information and are therefore redundant. This will help us narrow down the number of observables to subsets that can describe most of the information present in the dataset. Further, we investigate how the correlations are affected by the presence of the medium, that is, how they change between the Unquenched to the Quenched samples, to identify pairs of observables most sensitive to jet quenching in the QGP.

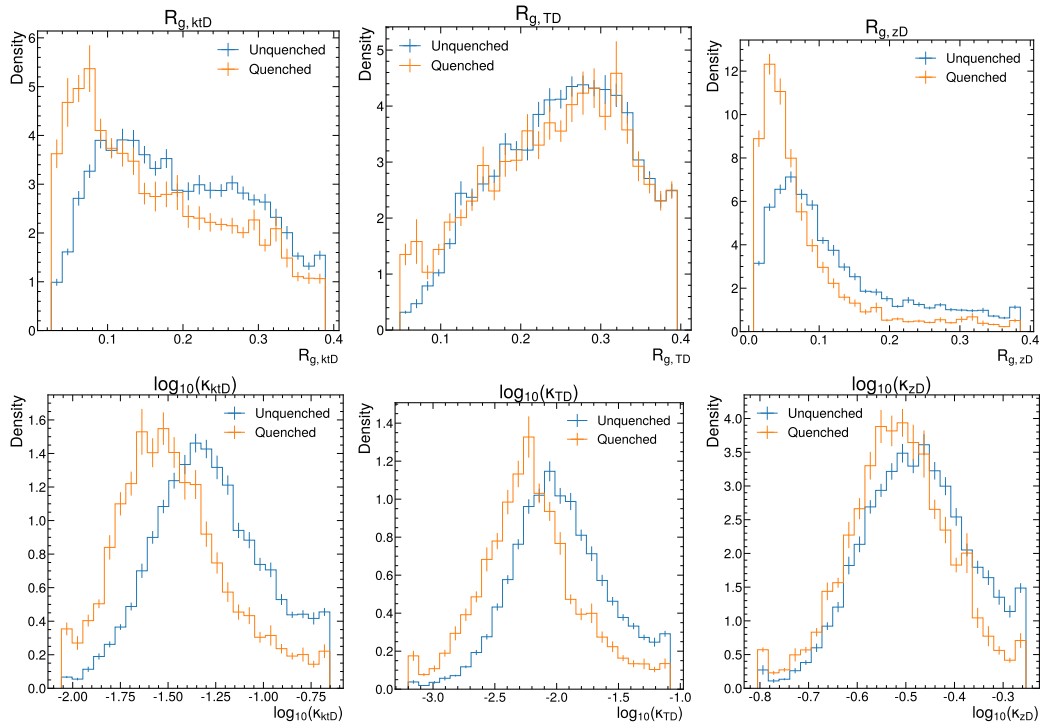

Figure 1: Distributions of dynamical grooming intrinsic observables produced by the three different dynamical grooming prescriptions: $p_T$-Drop (left column), TimeDrop (middle column) and $z$-Drop (right column).

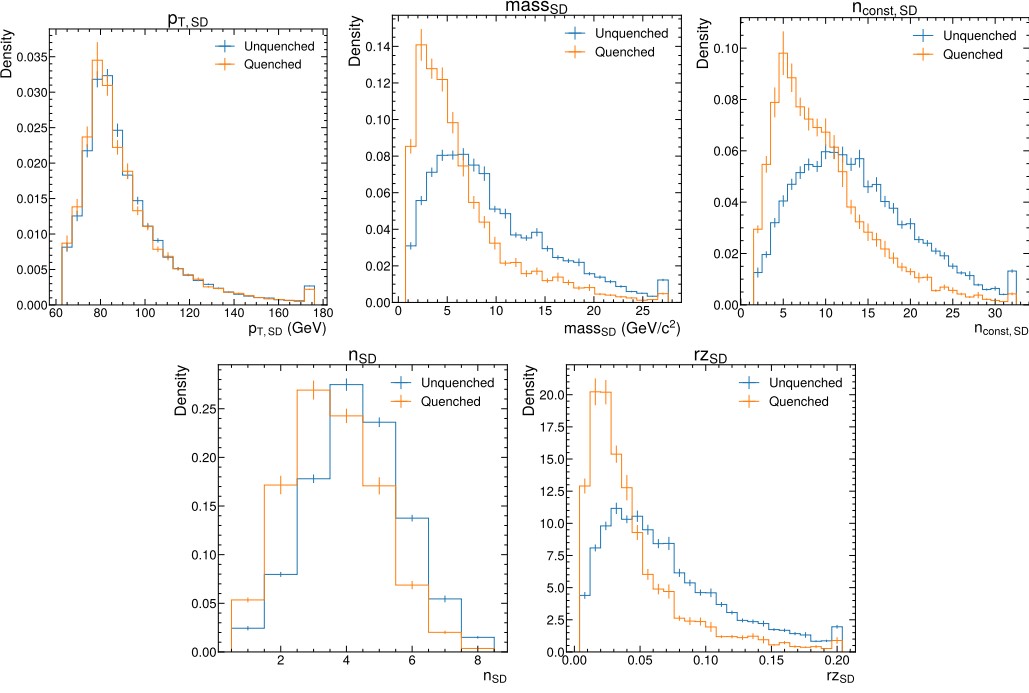

Figure 2: Example distributions of jet observables for jets with transverse momentum $p_T > 80\,\text{GeV}/c$ in the Quenched and Unquenched jet samples generated with JEWEL+ PYTHIA.

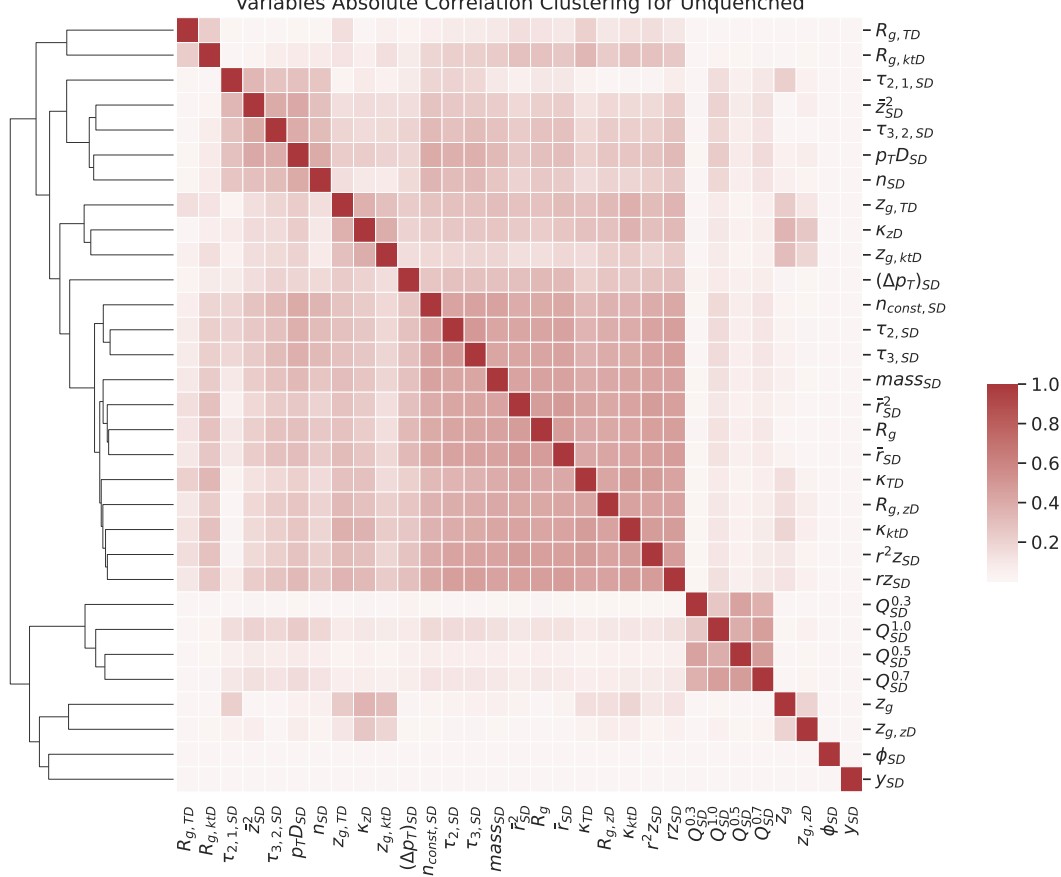

**Figure 3:** Clustermap for the Unquenched sample. The entries are the absolute values of the (Pearson's) correlation coefficients between two observables. The observables are reordered together in terms of the hierarchical clustering represented by the dendrogram in the left-side of the heatmap.

For each sample, Unquenched and Quenched, we compute the correlation matrix, i.e. the square matrix with all pairwise correlation coefficients across all observables, and take the absolute value of its entries as we are looking for relations in the data that do not need to be positive. To further illustrate these adjacency relations between observables, they have been clustered using hierarchical clustering. A visual representation of the correlation matrices and the result of the clustering for Unquenched and Quenched samples is shown in Figs. 3 and 4.

The hierarchical clustering used in Figs. 3 and 4 works very similarly to recursive jet clustering algorithms. Starting with the rows of the absolute covariance matrix (or columns, given its symmetry) compute all the euclidean pairwise distances between the rows. Then, cluster the two rows which are closer to each other, i.e. that have the smallest Euclidean pairwise distance, remove them from the set of rows and replace them with their average. This effectively reduces the number of rows by one. Recompute all pairwise distances with the new row replacing the two rows that were clustered together. Repeat until only one row is left. Here, the clustering is performed by `scipy`'s `hiearchy.linkage` function, using the `average` method which joins clustering branches by taking the average. Once this is done, one can set a threshold on the cluster distance to identify the main branches of the clustering, this is shown in Fig. 5. Both the threshold and the resulting main branches are arbitrary, but provide a visual guide on how alike the observables are.

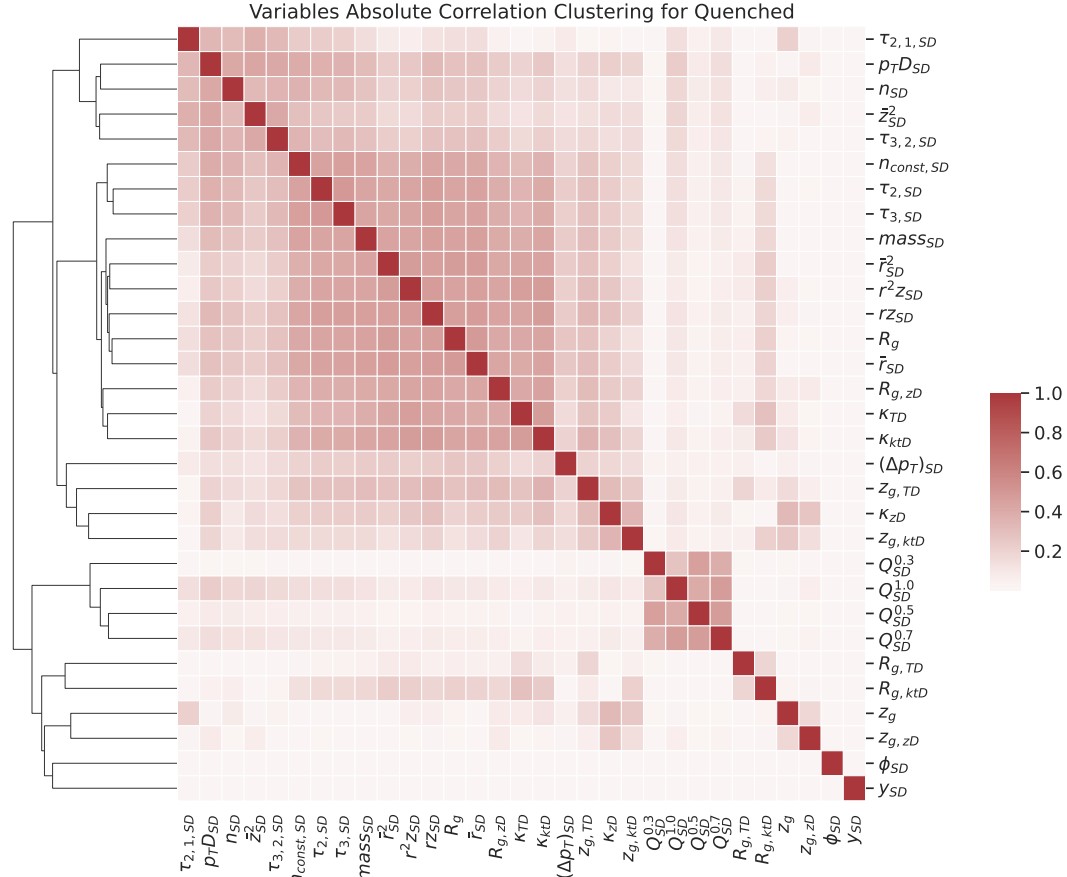

Figure 4: Clustermap for the Quenched sample. The entries are the absolute values of the (Pearson's) correlation coefficient between two observables. The observables are reordered together in terms of the hierarchical clustering represented by the dendrogram in the left-side of the heatmap.

In both Unquenched and Quenched cases, we observe that a very large group of observables are mutually correlated and clustered together in the dendrograms in Figs. 3 and 4 (as well in the coloured version of the same dendrograms in Fig. 5). This large cluster includes, in both cases, observables that capture the transverse substructure of the jet, like the angularities.

Comparing Figs. 3 and 4 (together with Fig. 5), one notable change is that in the sample with medium effects, some of the dynamical grooming observables ($\kappa_{zD}$, $z_{g,ktD}$, $z_{g,TD}$) and the momentum removed by SoftDrop $\Delta p_{T,SD}$ form a separate cluster rather than belonging to the large cluster. A second clear feature is that jet charge observables are closely correlated among themselves but have minimal correlation with the remaining observables. This indicates that these observables encode information not captured by other observables, but that this information is not modified by quenching.

As expected, since jets are produced uniformly in azimuth, $\phi$ is an independent observable (uncorrelated with all other observables) in both samples. While some correlation of the jet rapidity $y$ with other observables could be expected since average jet $p_T$ decreases with increasing $|y|$ and thus substructure observables could be correlated with rapidity, the rapidity and $p_T$ ranges considered in this study are sufficiently small to guarantee that rapidity $y$ remains an independent observable in both Unquenched and Quenched cases.

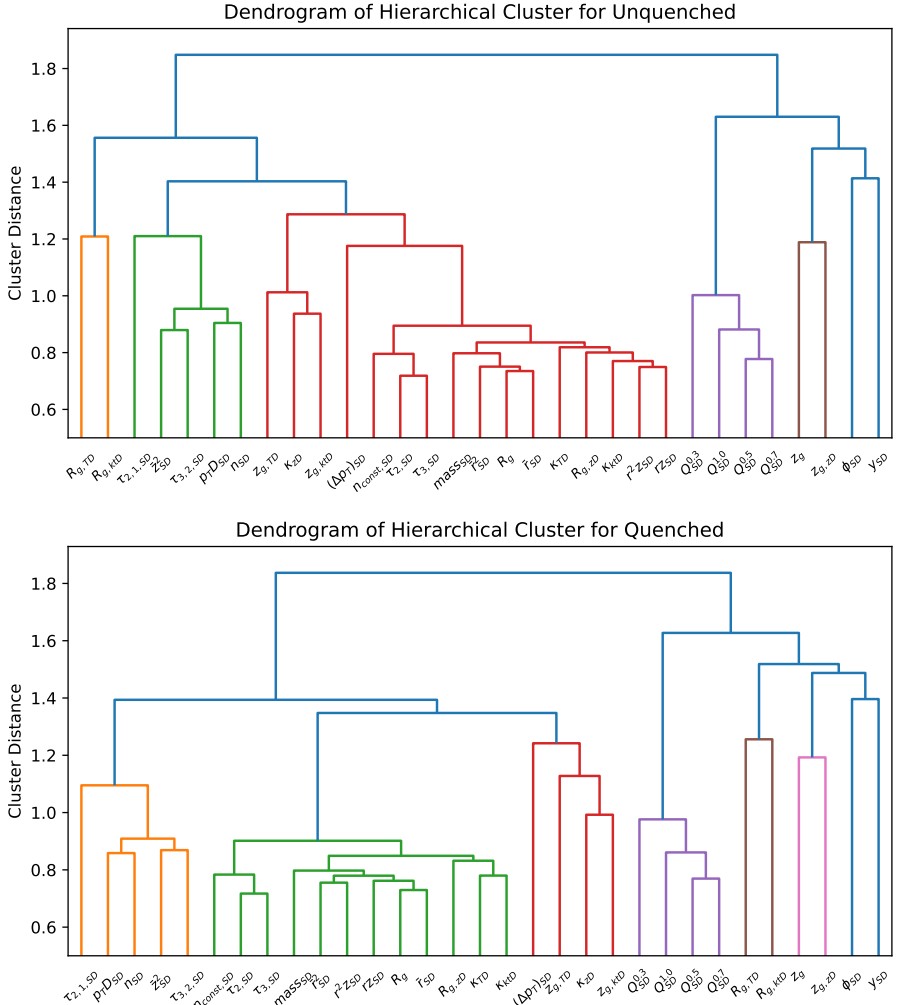

Figure 5: Dendrograms showing the clustering tree of the absolute values of the covariance matrices of Unquenched and Quenched samples. The colour threshold is set to 70% of the maximum cluster distance, above which a new colour is produced to represent a main branch. Given that in both samples the maximum distance between branches is just above 1.8, the main branches are identified as those that have clustering distance greater than around 1.3.

In order to highlight quenching effects we show, in Fig. 6, the difference between the absolute values of correlation coefficients in the Unquenched and Quenched samples for each pair of observables.

The most prominent feature in the change of the correlation strength is that the correlations of $(\Delta p_T)_{SD}$, $\tau_{2,1,SD}$, $R_{g,TD}$, and $R_{g,kTD}$ with most of the other observables are visibly different in the Quenched and Unquenched jet samples. Also the correlations of $\kappa_{zD}$, $z_g$, and $z_{g,ktD}$ with many other observables are significantly changed. In this part of the study, these observables are therefore identified as the most sensitive to quenching effects. The correlation of some pairs of observables change very significantly once a QGP medium is present.

To further understand how the observables are linearly correlated we performed a Principal Components Analysis (PCA). In PCA, we identify the main directions of the dataset, where by main direction we mean the direction over the observables basis that explains the most variance of the dataset. These main directions are called the principal components of the dataset.

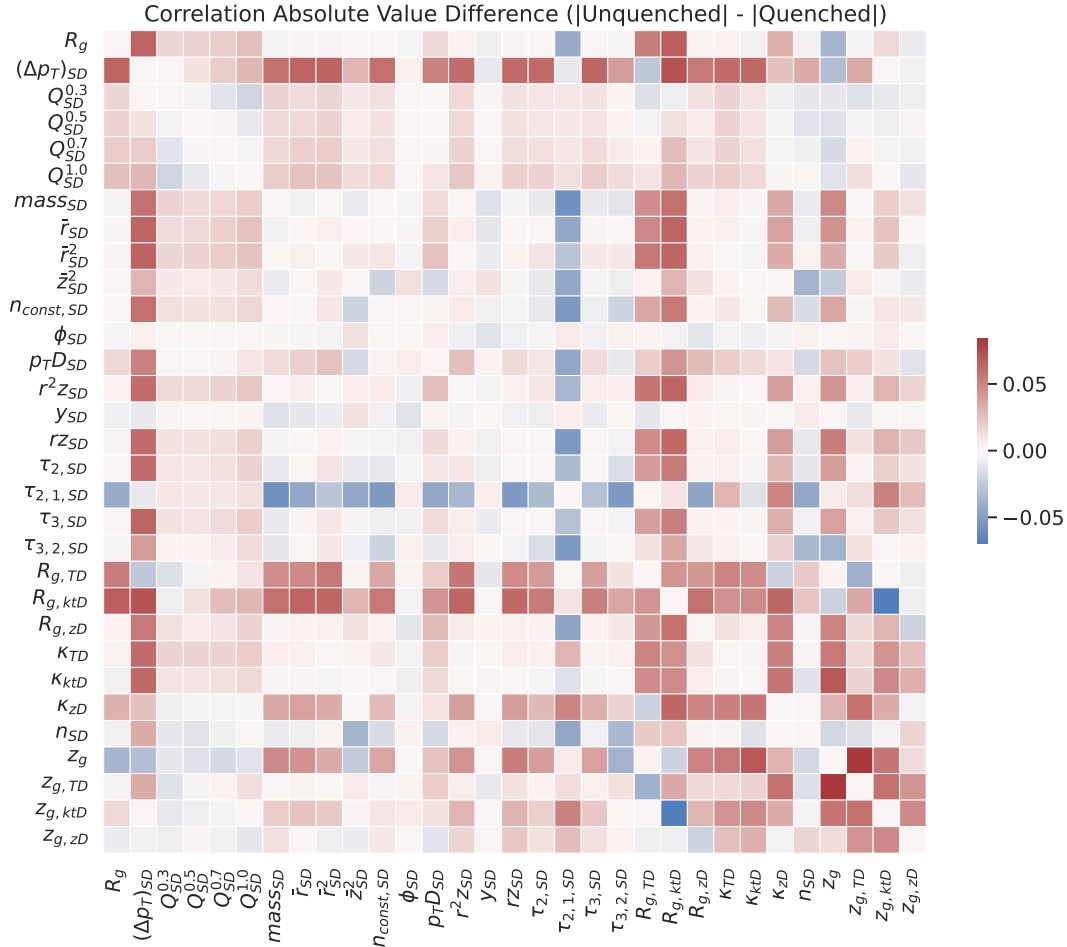

Figure 6: Difference of the absolute correlation matrices of the Unquenched and Quenched samples. Red means that the absolute correlation coefficient is smaller for the Quenched sample, while Blue means that the absolute correlation coefficient is larger.

Formally, the principal components are the column vectors that can be stacked together to form a rectangular matrix, $\mathbf{V}$, from which an orthogonal bases rotation can be performed such that it minimises the reconstruction error

$$\min_{\mathbf{V}} \mathbb{E}[\|x - \mathbf{V} \cdot \mathbf{V}^T \cdot x\|^2], \tag{12}$$

where $x$ is a vector of the observables of jet $i$,[2] and $\mathbb{E}[x] = \sum_i^{N_{jet}} w_i x_i / \sum_i^{N_{jet}} w_i$ the weighted expected value taken over the entire (training) dataset accounting for JEWEL+PYTHIA event generation weights. When $\mathbf{V}$ is composed of a single vector, this vector can be shown to be proportional to the eigenvector with the largest eigenvalue of the covariance matrix of the dataset. When $\mathbf{V}$ is composed of $N$ vectors, it can be shown that each is parallel to one of the $N$ eigenvectors of the covariance matrix corresponding to the $N$ largest eigenvalues. Thus, one can see the principal components as the eigenvectors of the covariance matrix of the dataset and therefore they represent the directions that carry the most variance between observables.[3] The optimisation problem Eq. (12) is then the problem of finding $\mathbf{V}$ that minimises the variance

---

[2]These are set to have vanishing expected value by normalising the dataset so that each observable has vanishing mean and unit variance using Scikit-Learn's StandardScaler [90].

[3]The PCA projection into $N$ principal components is therefore a dimensionality reduction algorithm. This has

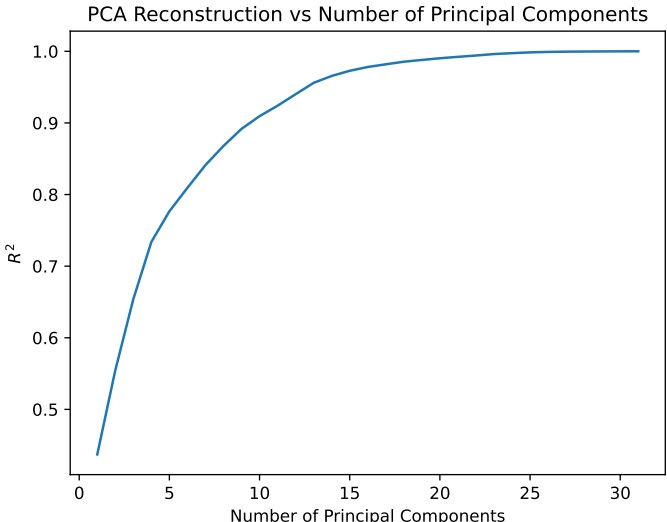

Figure 7: Coefficient of determination $R^2$ on the reconstructed data set after being rotated onto the principal components and back to the original base as a function of the number of principal components.

of the dataset in the principal components basis, which means that the principal components are themselves mutually linearly uncorrelated, i.e. orthogonal between themselves. Indeed, in the limit that $N$ is equal to the number of observables, then $\mathbf{V}$ is itself an orthogonal matrix composed of the all the eigenvectors of the covariance matrix and the reconstruction error is trivially vanishing as $\mathbf{V}\cdot\mathbf{V}^T = \mathbf{1}$. We perform PCA on the Unquenched training set. This defines the principal components of this sample, which will allow us to study how the Unquenched PCA rotation affects the Unquenched and Quenched samples. The implementation of weighted PCA, where the event generation weights were used to derive the statistics that produce the covariance matrix, was performed using the Python package `wpca` [92].

To assess how well the principal components capture the linear relations between the observables, we compute the coefficient of determination, $R^2$, which quantifies the quality of the reconstruction after performing a rotation by $\mathbf{V}\cdot\mathbf{V}^T$,

$$R^2(x,\hat{x}) = 1 - \frac{\mathbb{E}[\|x - \hat{x}\|^2]}{\mathbb{E}[\|x - \mathbb{E}[x]\|^2]}, \tag{13}$$

where $x$ are the observables vectors, $\hat{x} = \mathbf{V}\cdot\mathbf{V}^T\cdot x$ is the reconstructed $x$ after the being rotated into the principal components and back. This metric measures how well the observables are reconstructed after being projected into the principal components and back to the original basis, therefore quantifying how much information was retained. It usually takes values between 0, for a baseline where the reconstruction simply reproduces the average value of the observable for all jets ($\hat{x} = \mathbb{E}[x], \; \forall_x$), and up to 1 when the value of the observable for each jet is reproduced accurately, amounting to perfect reconstruction. It is however possible to obtain negative values for $R^2$ when not even the mean value of the observable is reproduced. In this cases, the second term in Eq. (13) becomes very negative if the reconstruction is very different from the actual value or its mean. Intuitively, the coefficient of determination can be seen as a normalised Mean Square Error, as the numerator of the fraction is the sum of all square errors and the denominator is the sum of all residuals. This fraction also often takes

---

been explored for dataset dimensional reduction in HEP in [91] in the context of quantum machine learning in current and near future quantum computers.

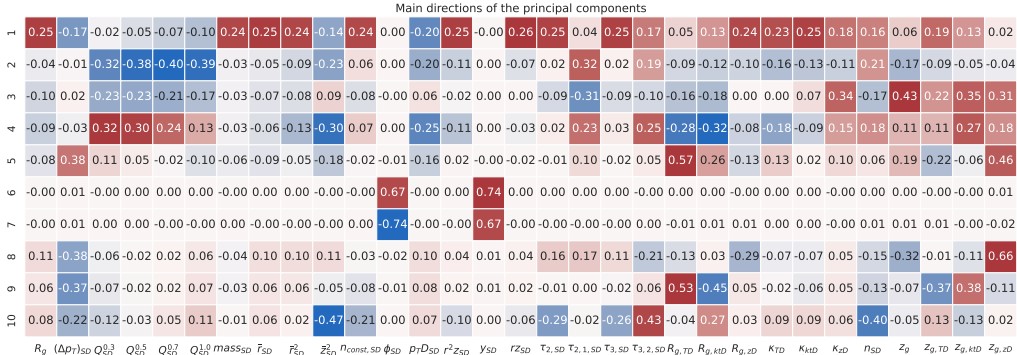

Figure 8: Distribution of the first 10 main principal components. Each component has unit vector norm. The values can be interpreted as weights for each observable in each principal component: larger absolute values mean a larger contribution of the observable.

the name of Fraction of Variance Unexplained, which reinforces the notion that $R^2$ is capturing the fraction of the variance that is being encoded into the principal components.

In Fig. 7 we show how $R^2$ increases as we increase the number of principal components. This increase is such that the first up to 10 components are driving most of the linear relations between the observables, and by the tenth component we are already capturing around 90% of the variance with a simple orthogonal rotation. This result will be compared later in this work with an analogous result obtained from considering non-linear maps to study the relations between observables.

In Fig. 8 we present a visual representation of the first ten components by representing the contribution of each observable with a colour scale. We observe the same trend as in the clustermaps. The first principal component is a combination of mostly the angularity-type observables, while the second principal component mostly involves the jet charges, which are strongly correlated amongst themselves. The third component captures some of the grooming observables, namely the $z_g$'s. The forth component appears to be disentangling the effect of the jet charges from some high-level observables, such as $\tau$ ratios and dynamical grooming $R$'s. This separation of observables in several groups is to some extent in line with expectations: for example the jet charges are explicitly sensitive to the total charge of the jet, and the charge of the parton producing the jet, while the angularities measure the momentum distributions inside jets. The grooming and subjettiness observables are expected to be more sensitive to specific substructure than the angularities. The fact that many observables within each category are grouped together in each principal component suggests that the number of independent degrees of freedom in the data set is much smaller than the number of observables that was studied. The interpretation becomes harder for higher (larger $N$) principal components since then we need to consider the directions already captured by the previous $N-1$ principal components, as the linear relations are already described by the preceding components and higher components begin to capture the tails and noise of the distributions. Around components six and seven, the PCA is finally learning the $\phi$ and $y$, which are uncorrelated to the rest of the observables. This means that most of the multi-observable correlations have already been captured in the first five principal components, and we therefore focus only on the first five components from hereon.

Whilst Fig. 7 shows how $R^2$ varies as we increase the number of principal components, it does not tell us how the individual observables are driving its value. In Fig. 9 we show how the contribution of each observable to the total $R^2$ changes as we increase the number of principal components. The figure illustrates how the variance related to each observable[4] is

---

[4]Quantified here by the reconstruction quality, $R^2$, under the PCA rotations for each value of the number of

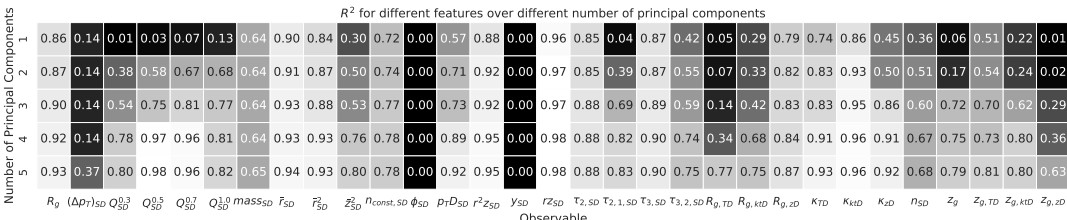

Figure 9: Reconstruction quality, $R^2$ (c.f. Eq. (13)), per observable as a function of the number of principal component on the Unquenched sample.

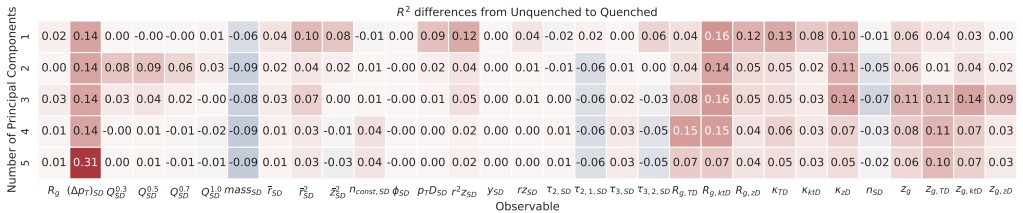

Figure 10: Difference in the quality of reconstruction between the Unquenched and Quenched samples using the PCA derived from the Unquenched sample.

explained by the principal components. We see how with a single component the angularity-type observables are reasonably reconstructed, implying that a single degree of freedom was able to capture most of $R_g$, $\bar{r}_{SD}$, $\bar{r}^2_{SD}$, $p_T D_{SD}$, $r z_{SD}$, and some of the dynamical grooming $\kappa$ observables.

In order to explore the impact that jet quenching can have in different observables and their correlations, Fig. 10 shows how $R^2$ changes when using the principal components rotation computed in the Unquenched sample to rotate the Quenched sample. These numbers indicate changes in the relations between different observables in the Unquenched and Quenched samples; larger (absolute) values indicate that the shape of given observable in the Quenched sample deviates from the pattern in pp, suggesting a larger medium effect on the observable. We observe that $(\Delta p_T)_{SD}$, and the dynamical grooming observables, in particular the $\kappa$s and subjet distances $R_g$, show the large differences between the Unquenched and Quenched samples. In addition, we observe large values for angularities $\bar{r}^2$ and $r^2 z$. These observables are candidates for identifying jet quenching effects. We will return to this discussion later in Section 6.

In Fig. 11 we present the distribution of the first six principal components for the Unquenched and Quenched samples. A clear difference between Unquenched and Quenched is visible in the projection on the first principal component, which has contributions from most of the angularity-type observables. The second principal component, which is mostly aligned along the jet charges, is not sensitive to quenching effects. The remaining components show a weak sensitivity to jet quenching.

With this analysis we learnt that the angularities, as well as the N-subjettiness and some of the subjet angles $R_g$, are highly linearly correlated with each other. In fact, a single principal component appears to be capturing most the information of these observables. This component also presents some discriminating power between Unquenched and Quenched samples, which we will explore in more detail below. The principal component analysis only captures linear relations between observabless and fails to capture further non-linear relations, a shortcoming that is discussed and tackled in the next section.

---

principal components.

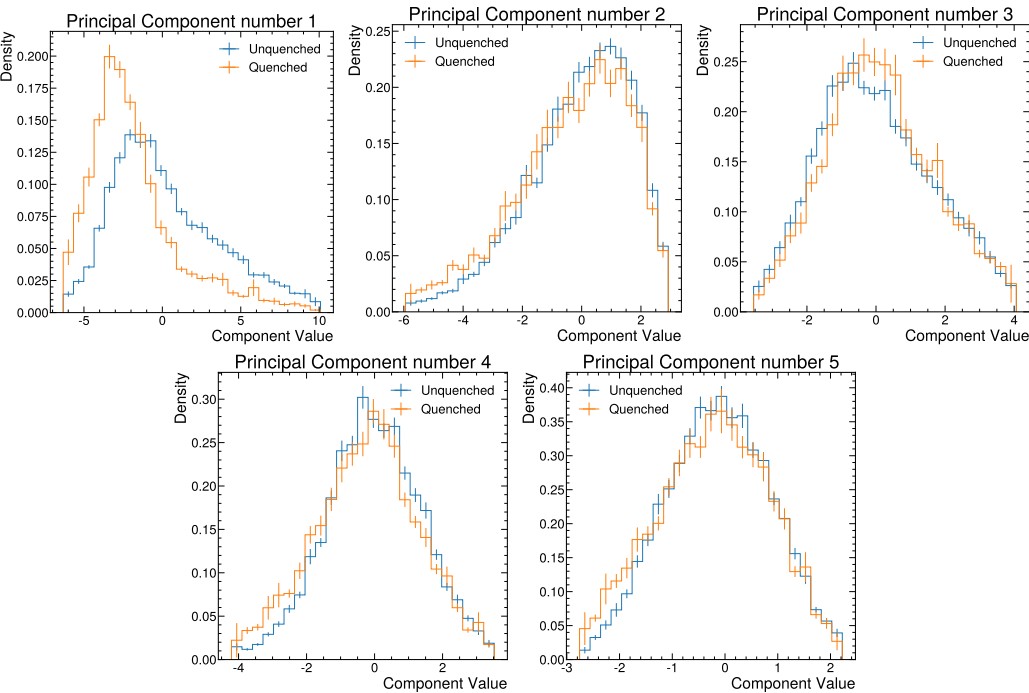

Figure 11: The distribution of the first five principal components (using the transformation from the Unquenched sample) in the Unquenched and Quenched test sets.

## 5 Deep auto-encoder analysis

One of the limitations of the PCA analysis presented in Section 4 is that it only captures linear relations amongst observables since it uses rotations of the basis vectors of the covariance matrix. In this section we will make use of a Deep Auto-Encoder, which will provide an analogous discussion to that presented before, while also capturing non-linear relations between observables. Deep Auto-Encoders have been explored in HEP in the context of Anomaly-Detection in searches for new physics [93–96], while here we will use them as a tool for data analysis.

A Deep Auto-Encoder, $AE$, is a neural network architecture that attempts to minimise a loss function analogous to Eq. (12), i.e. attempt to reconstruct the inputs as they are fed-forward through the network, using a neural network with a bottleneck layer with a size much smaller than the number of observables. This means that the $AE$ learns how to project the data into a lower dimensional space, i.e. to encode it, and then to reconstruct the inputs back to their original form, i.e. to decode it. This hidden layer is usually referred to as the latent space, $z$, which can have any dimension, $z_{dim}$.[5] A diagram of a Deep Auto-Encoder neural network structure can be seen in Fig. 12.

The loss function used to train the $AE$ is very similar to the one in the PCA, but instead of finding the optimal orthogonal rotation, we want to find the optimal non-linear map implicit in the $AE$

$$\min_{\mathbf{w}} \mathbb{E}[\|x - AE(x, \mathbf{w})\|^2], \tag{14}$$

where $w$ are the trainable parameters of the neural network, $AE$, and $x$ are the inputs, i.e. the data.

The dimension of the latent space in the $AE$ plays a similar role as the number of principal

---

[5]This is the customary notation for latent space in deep learning architectures, not to be confused with the $z$ jet fragmentation fraction observables. For the remaining of the text, only the dimension of the latent space, $z_{dim}$, is of interest, so there should be no confusion.

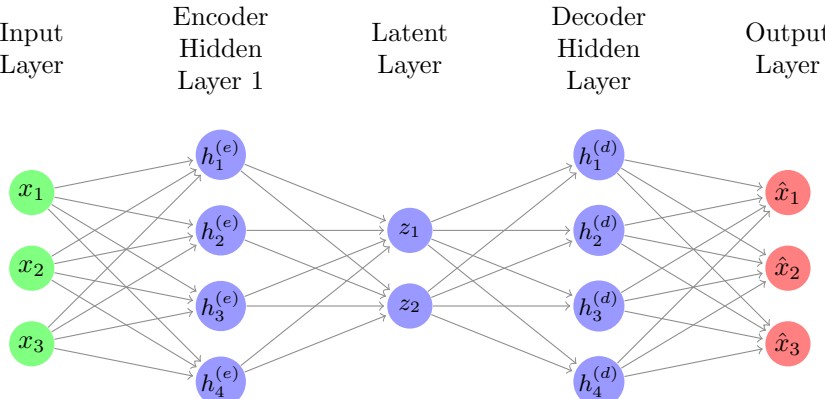

Figure 12: Deep Auto-Encoder schematic. In this schematic, the data has three observables, both the encoder and the decoder have only one hidden layer with four nodes, and the latent space has dimension equal to two.

components in the PCA. In the PCA, the rotation maximises the amount of variance the first principal components explain, capturing the most relevant mutual linear correlations. Likewise, we expect the *AE* to be able to capture the non-linear relations that explain the largest group of correlated observables at lower $z_{dim}$, and progressively start explaining more subtle effects (and even noise) as we increase the number of $z_{dim}$. In the limit that $z_{dim}$ equals the number of observables, the *AE* model will approach the identity function, obtaining perfect reconstruction without learning any relations. Fig. 13 shows the evolution of $R^2$ for increasing number of hidden dimensions $z$ in the *AE* in comparison to its evolution for increasing number of principal components in the PCA analysis of Section 4. Beside $z_{dim}$, which is the main parameter of interest in this analysis, there are a number of hyperparameters that determine the training process of the *AE* which need to be chosen: the number of the encoder and decoder layers, their width (i.e. the number of nodes), the non-linear activation function, and optimisation details. Choosing the optimal combination of such parameters can be difficult when performed manually. We optimised the model hyperparameters using the python package `optuna` [97]. The network itself was implemented using `TensorFlow` [98], using its high-level API, `Keras` [99]. The hyperparameter space and optimisation details can be found in Appendix B.

The hyperparameters are tuned for each value of the $z_{dim}$, in order to maximise the quality of the *AE* reconstruction, i.e. to maximise $R^2$, c.f. Eq. (13). The value of the $R^2$ for the best *AE* for each hidden latent space dimension is shown in Fig. 14, where by comparing with the analogous quantity from the PCA we see that the *AE* reproduces the data better for lower $z_{dim}$ than the PCA at the same number of components, which is due to the *AE* capacity to learn non-linear relations present in the data.

In Fig. 14 we present the value of $R^2$ per observable as we increase $z_{dim}$, where we restrict to the first five dimensions as we have learned from the PCA analysis that these are the most interesting ones. We see that with only one dimension, the *AE* learns the basic relations between what we have been calling angularity-type observables; most of the other observables are not described well. This suggests, just like in the PCA study, that the angularity-type observables are strongly related to each other, although here (as opposed to the PCA case) we are capturing non-linear relations as well as linear relations. As $z_{dim}$ increases, the *AE* unsurprisingly performs progressively better in encoding and decoding the rest of the observables.

Since the *AE* was trained on unquenched jets, we can study how it performs when presented with Quenched samples. The change of the performance, measured in terms of $R^2$ for each observable, due to quenching effects is presented in Fig. 15. Just like the analogue discus-

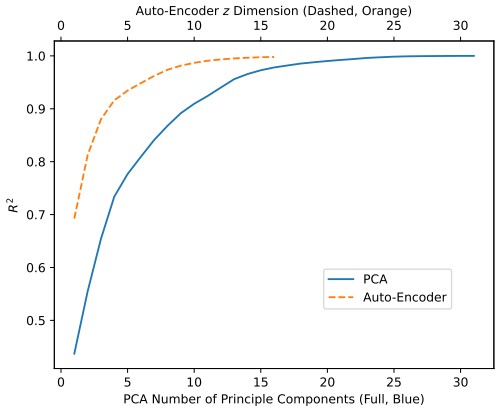

Figure 13: Quality of reconstruction, $R^2$ (c.f. Eq. (13)), as a function of the number of latent dimensions, $z_{dim}$, in the deep autoencoder (orange dashed line). The result with the PCA method as a function of the number of principal components is shown for comparison (blue line).

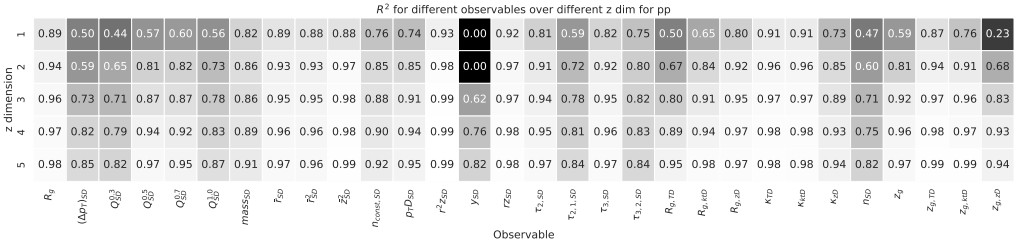

Figure 14: Contribution of each observable (columns) to the explained variance $R^2$ as a function of the number of latent dimensions (rows).

sion in the PCA section, higher (absolute) values reflect changes to the patterns and relations of the observables. Here we can see that the observables for which the reconstruction changes the most due to the presence of the medium are the subjet distances $R_g$ from the $k_T$- and time-ordered dynamical grooming, $(\Delta p_T)_{SD}$, and the softdrop $z_g$. In addition, $\kappa_{zD}$ from the $z$-based dynamical grooming, the number of softdrop splittings $n_{SD}$, and the n-Subjettiness ratio $\tau_{2,1}$ are also affected significantly. Also here, the changes in the description of the observables in the quenched sample are only sizable for a small number of dimensions.

It is also interesting to note that already for $z_{dim} = 5$, the $R^2$ differences in Fig. 15 become very small, i.e. the autoencoder that is trained on Unquenched events provides a very accurate prediction also for Quenched events. This suggests that the relations between some of the observables are very similar in quenched and unquenched jets, even if the mean values for specific observables may change due to quenching. This is further explored in Section 6 (see in particular Fig. 19).

With these two analyses, we have identified that the dynamical grooming observables possess information which is not included in the angularity-type observables, and that such information is relevant for the discrimination between unmodified and quenched jets. In the next analysis we will focus more on understanding which observables are most sensitive to jet quenching effects.

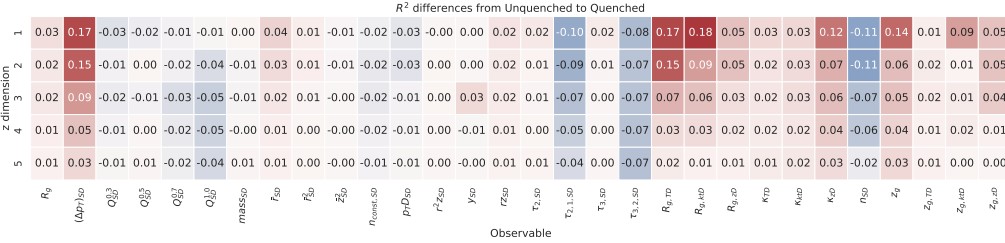

Figure 15: The change of $R^2$ contributions for each observable when using the auto-encoder that was trained on events without quenching to predict values for the quenched events.

## 6 Unquenched vs quenched discrimination analysis

In the previous two analyses, we have used PCA and Deep Auto-Encoders to identify observables which are related, i.e. that share the same underlying information. We have also shown how these results are affected by jet quenching in the QGP. In this section we will focus on the sensitivity of each observable, and of pairwise combinations of observables, to jet quenching effects. To do so, we will be exploring Boosted Decision Trees (BDTs) as implemented by the python package xGBoost [100] to distinguish jets from the Unquenched and Quenched samples. As we want to learn about the sensitivity of each observable to medium effects, we create a strong baseline by training a BDT using all observables. The output of this BDT and its performance on the test set is shown in Fig. 16. Apart from statistical fluctuations, this is the best possible discrimination between jets in the Quenched and Unquenched samples that we can expect using all the presented observables. We note that this discrimination is complicated by the fact that the Quenched sample also contains jets that experienced little of no modification by the QGP and, as such, are indistinguishable from those in the Unquenched sample. In the same figure we also show the receiver operating characteristic (ROC), which is obtained by plotting the true positive rate (TPR) against the false positive rate (FPR) at different classification thresholds on the BDT output, where

$$TPR = \frac{TP}{TP + FN}, \tag{15}$$

$$FPR = \frac{FP}{FN + TP}, \tag{16}$$

where $TP$ stands for the counts of true positives, i.e. quenched jets which are identified as such, $FN$ the counts of false negatives, i.e. quenched jets which are incorrectly identified as unquenched, and $FP$ the counts of a false positives, i.e. unquenched jets which are incorrectly identified as quenched, all computed at a fixed threshold, i.e. at a specific cut on the BDT output. Alternatively, using HEP nomenclature, one could also have called *TPR Quenched Jet Efficiency*, and $1 - FPR$ *Unquenched Jet Rejection*. The area under the curve (AUC) of the ROC therefore represents how well a classifier performs for different operating thresholds of the cut, with a perfect classifier having a ROC AUC equal to 1 and a random classifier 0.5, corresponding (respectively) to a lack of overlap of the classifier output for both classes, or a complete overlap. On the right-hand plot of Fig. 16 we see that the BDT over all observables has a ROC AUC of 0.701, which means that while it performs better than a random classifier, the BDT output distributions of both classes have a considerable overlap, as can be seen on the left-hand plot of Fig. 16. Further below we will study the classification task for an explicit operating point of 0.5 Quenched Jet Efficiency.[6]

---

[6] In HEP, the accuracy metric, i.e. the fraction of true classifications at the operating point corresponding to cutting at 0.5 of the BDT output, is often not helpful as one strives to optimise the statistical significance of the

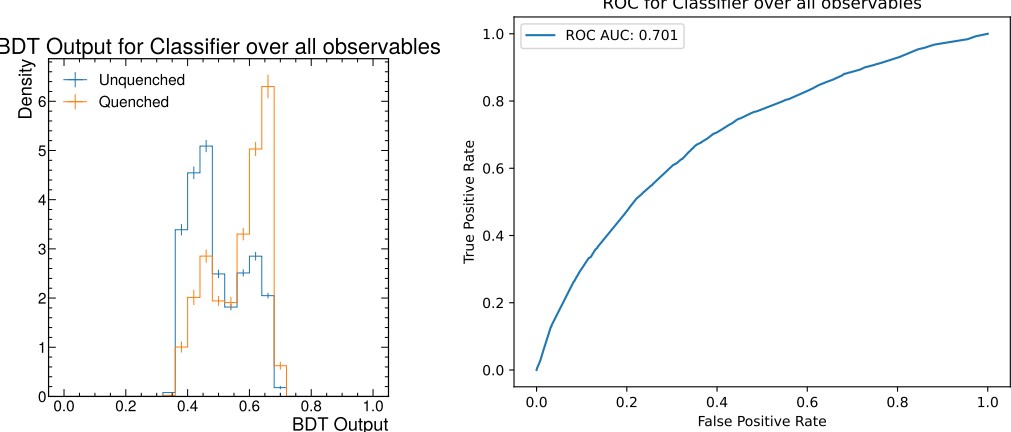

Figure 16: Left: Distribution of the output of the BDT on the test set. Right: receiver operating characteristic (ROC) curve of the BDT.

Having a strong baseline for discrimination performance, we then separately train BDTs over each single observable and over each pair of observables. The goal of this analysis is to identify a small set of observables that are most sensitive to medium-induced modifications, to avoid the need to use high-level multivariate classifiers, such as a BDT or a neural network classifier, in data analysis. To do this, we measure the discriminating performance of each of these BDTs by calculating the ROC AUC and compare this value to the ROC AUC of the BDT trained using all the observables.

In Fig. 17 we present a heatmap of the AUC ROCs for all combinations, normalised to the value obtained by the BDT over all observables, i.e. 0.701. It is noteworthy that some observables are sensitive to QGP effects by themselves: $rz_{SD}$ and $\tau_{2,SD}$ accounting individually for 0.99 of the discriminating power of the BDT trained in all observables, and $n_{\mathrm{const,SD}}$, $r^2z_{SD}$, $\tau_{3,SD}$, $\kappa_{ktD}$ with 0.98. Of particular importance are the pairs of observables that saturate the discrimination power of the BDT trained in all observables. These are pairings of $rz_{SD}$ with $(\Delta p_T)_{SD}$, $\tau_{3,SD}$, $\kappa_{TD}$ or $\kappa_{ktD}$; also the further pairings of $\kappa_{ktD}$ with any of $n_{\mathrm{const,SD}}$, $p_T D_{SD}$, $\bar{z}^2_{SD}$, $\tau_{2,SD}$ or $\tau_{3,SD}$: and $\kappa_{TD}$ with $\tau_{2,SD}$. Consistently with our previous discussion, pairs involving a dynamical grooming observable and an angularity-type observable dominate this list.

To further illustrate the sensitivity to quenching of the observables identified above, we focus on $\kappa_{ktD}$, $rz_{SD}$, and $\tau_{2,SD}$. To assess the discrimination power for each of these observables, we find the cut value at which half of Quenched sample is accepted and calculate the rejection of events from the Unquenched sample for each cut. Table 2 shows the cut values and the corresponding Unquenched rejection efficiency.

This study shows that $\kappa_{k_{TD}}$, $rz_{SD}$, and $\tau_{2,SD}$ have similar discriminating power when used as a taggers for jet quenching. In Fig. 18 we show how cutting in each of these observables affects the distribution of the remaining observables. We observe that all the cuts seem to be capturing a similar subsample of jets, as the post-cut distributions (full lines) are similar across all selections.

In Fig. 19 we present the 2D distributions, and their difference between the Quenched and Unquenched samples, for some of the most sensitive observable pairs. The red and blue areas in each panel show where the population is larger in the Unquenched and Quenched samples respectively. The figure shows that the relation between the different selected observables are very similar in the Quenched and Unquenched samples, but that the most probable value is different between the two samples. For example, in the middle panels, the relation between

---

class of interest, i.e. the optimal trade-off between efficiency and purity depends is analysis-depend.

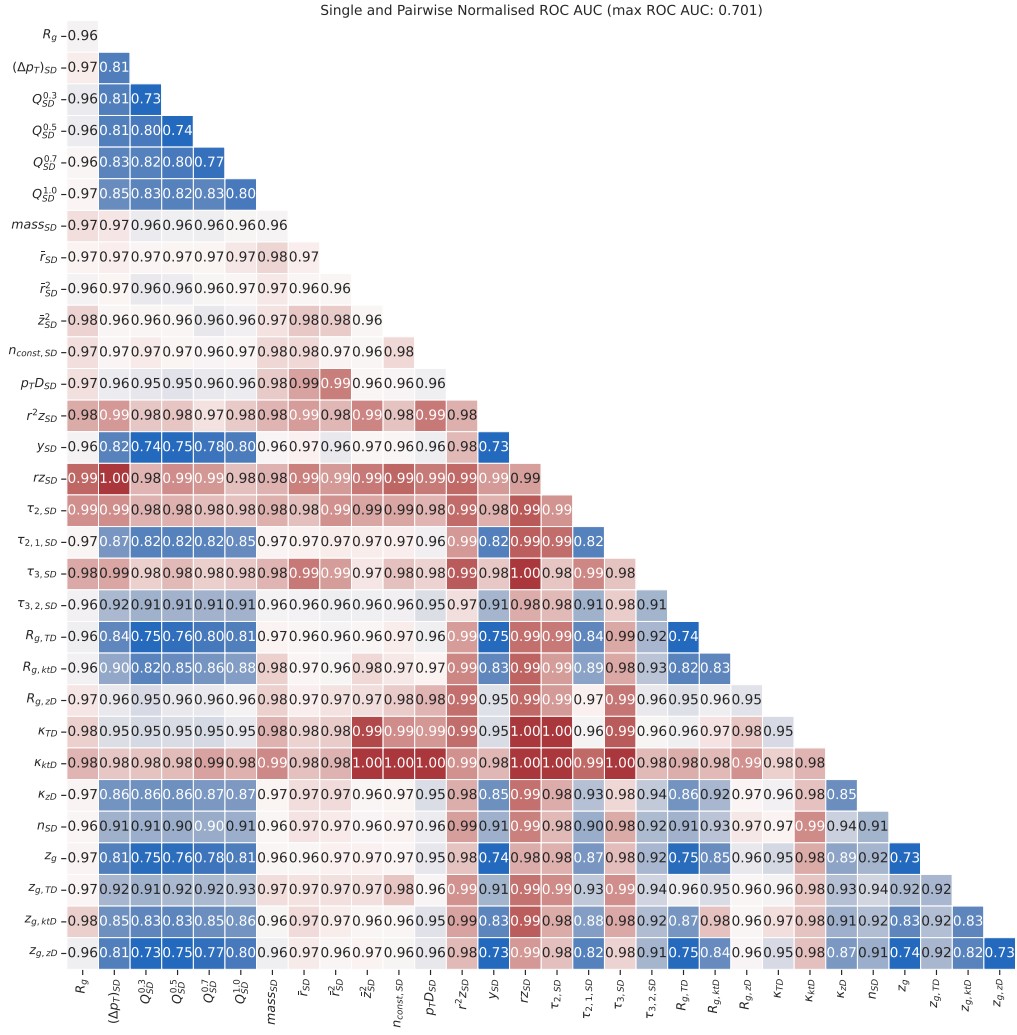

Figure 17: Single and pairwise ROC AUC normalised to the ROC AUC obtained using all observables. Values close to 1 signify that almost all discrimination power of the full BDT is being captured.

$rz_{SD}$ and $\log_1 0(\kappa_{kTD})$ is very similar for the Quenched and the Unquenched sample, and the effect of quenching is a shift of the population to smaller values of both observables. For both top and middle panels the resilience of the correlation to quenching effects was already present for linear correlations as identified in our PCA analysis (see Fig. 6) which is consistent with the approximate linearity of the correlation seen in Fig. 19. The case depicted in the

Table 2: Cuts over selected observables at 0.5 Quenched Acceptance Efficiency (True Positive Rate) for quenched jets from the JEWEL+PYTHIA event generator with Quenched settings and their respective Unquenched Rejection Efficiency (True Negative Rate).

| Observable | Cut | Unquenched Rejection Efficiency |
|---|---|---|
| BDT Output | $> 0.59$ | 0.78 |
| $\kappa_{k_{TD}}$ | $< 0.03$ | 0.78 |
| $rz_{SD}$ | $< 0.03$ | 0.78 |
| $\tau_{2,SD}$ | $< 0.04$ | 0.77 |



Figure 18: Distributions after the different cuts. Each row shows the distributions without (solid line) and with (dashed line) selection for Quenched (orange) and Unquenched (blue) JEWEL+PYTHIA events. The selection observable is different for each row of panels. Inclusive distributions are normalised to unit area. The cut distributions for the quenched sample are normalised to 0.5 area (the acceptance efficiency). Unquenched cut distributions are normalised to appropriate area, i.e. one minus the Unquenched rejection efficiency.

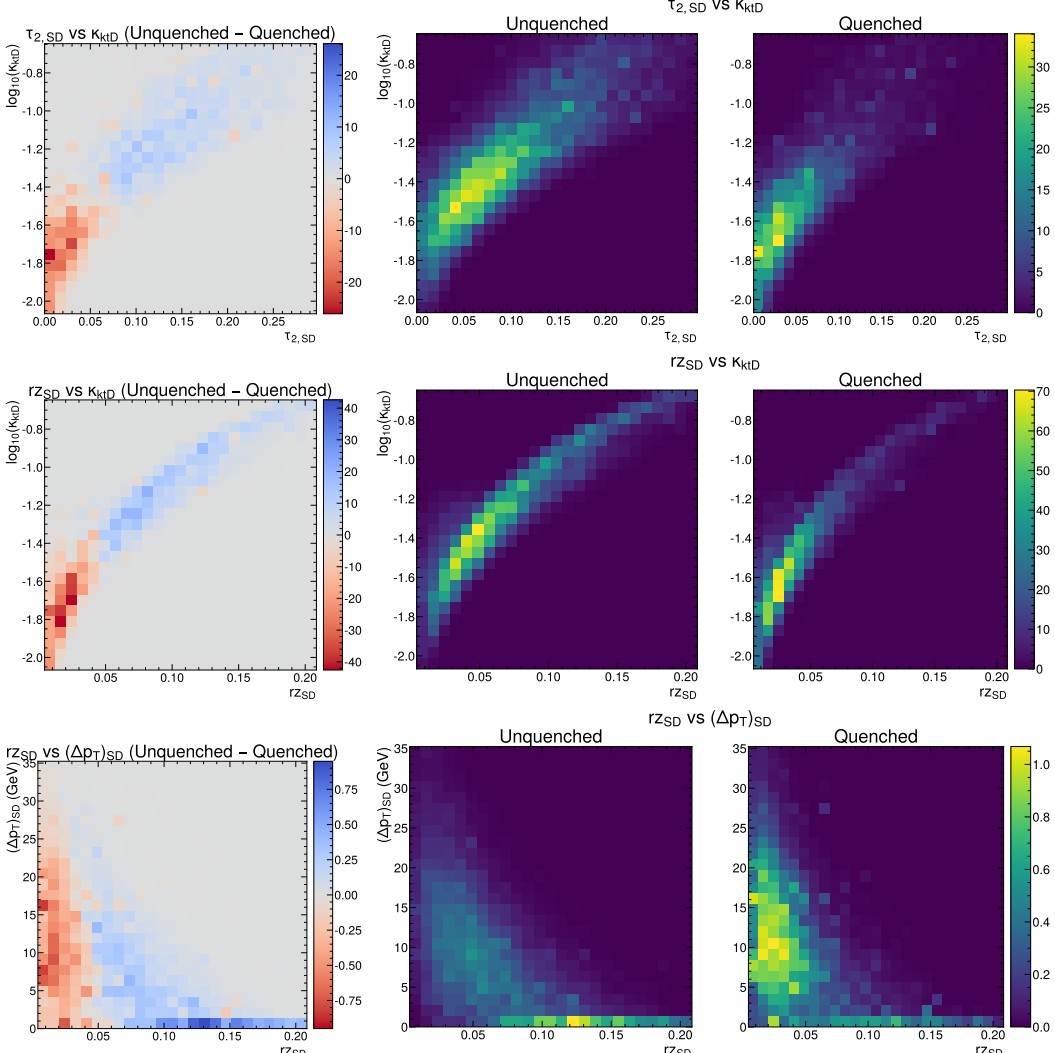

Figure 19: Left: Difference between the Unquenched and Quenched two dimensional densities across some of the most medium sensitive pairs. Blue (Red) means that the density is greater for Unquenched (Quenched). Right: Densities for each sample for the same pair of observables.

lower panel is different. The linear correlation, captured by the PCA analysis, between $rz_{SD}$ and $(\Delta p_T)_{SD}$ is not strong in either Unquenched or Quenched samples (see Figs. 3 and 4) and is strongly modified by quenching (see Fig. 6). However, the ability of the AE ot capture non-linear relations between these observables makes their (non-linear) correlation resilient to quenching effects. Again here, quenching effects result in a strong population migration, in this case for low values of $rz_{SD}$.

In this analysis we explored the relations between observables by using the discriminating power between Unquenched and Quenched samples to determine their sensitivity to medium effects. We found that selected pairs, mostly involving angularity-type observables, such as $rz_{SD}$, in combination with the higher-level observables $\kappa$ obtained from the dynamical grooming procedure, provide a discriminating performance close to that obtained by using all the considered observables. We also showed that when using a selection that rejects 50% of the Quenched jets using these observables, the obtained rejection efficiency for Unquenched jets is similar to that of the full BDT, and that the rejection power is similar for each of the observables

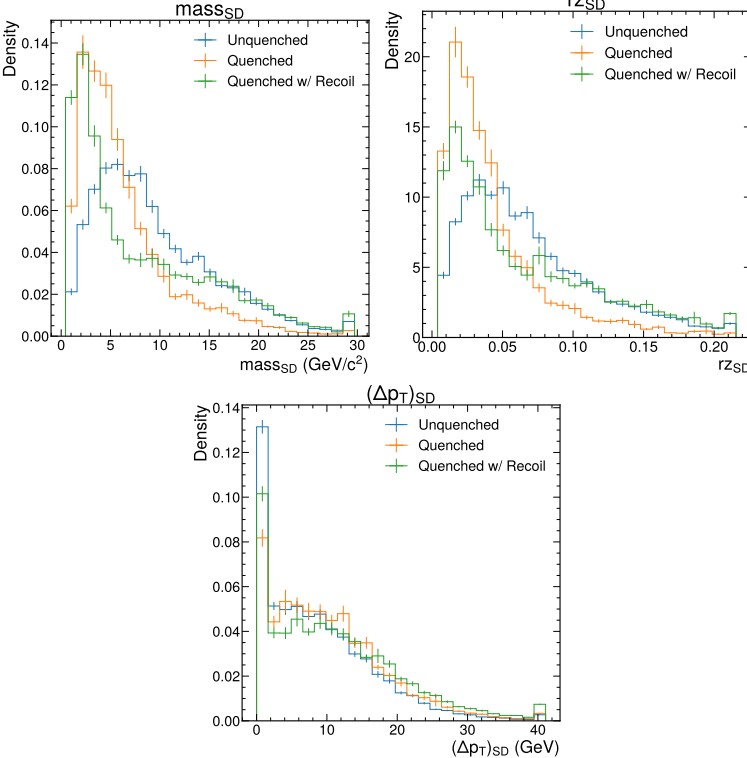

Figure 20: Distributions of SoftDrop jet mass, jet girth $rz_{SD}$, and the change of transverse momentum $(\Delta p_T)_{SD}$ from JEWEL+PYTHIA without quenching (blue lines) and with quenching, without (orange) and with recoil enabled (green). For the sample with recoil, the thermal background is subtracted using the constituent subtraction procedure from JEWEL [70].

in this set. This further suggests that the observables studied in this work are considerably correlated, and only even simple observables like $rz_{SD}$ can provide almost optimal discrimination between unquenched (vacuum) and quenched jets.

## 7  Impact of QGP response

The proof-of-principle analyses presented in the sections above were performed without considering the contribution of QGP response to the jets. These contributions, albeit a small contribution to the total transverse momentum reconstructed within a jet, can have significant effects on the jet substructure observables considered in this study [62–69] and are an important element in reaching agreement between JEWEL results and experimental measurements for some of those observables [69,70]. On general grounds it is expected that the inclusion of QGP response contributions to jets will increase the ability to discriminate strongly modified jets from those that suffered little or no modification (be it for jets produced in the absence of QGP where no modification can occur or jets that while developing within QGP suffered little modification). This is so for two reasons: QGP response is a feature obviously only present in jets that propagated in the QGP, and the effect is larger for jets which deposit more energy-momentum in the QGP, which are also expected to show stronger modification. This is shown explicitly in for example [72].

A related but separate issue is how robust Machine Learning based discrimination strategies are against the inevitable contamination of experimentally reconstructed jets by the large and

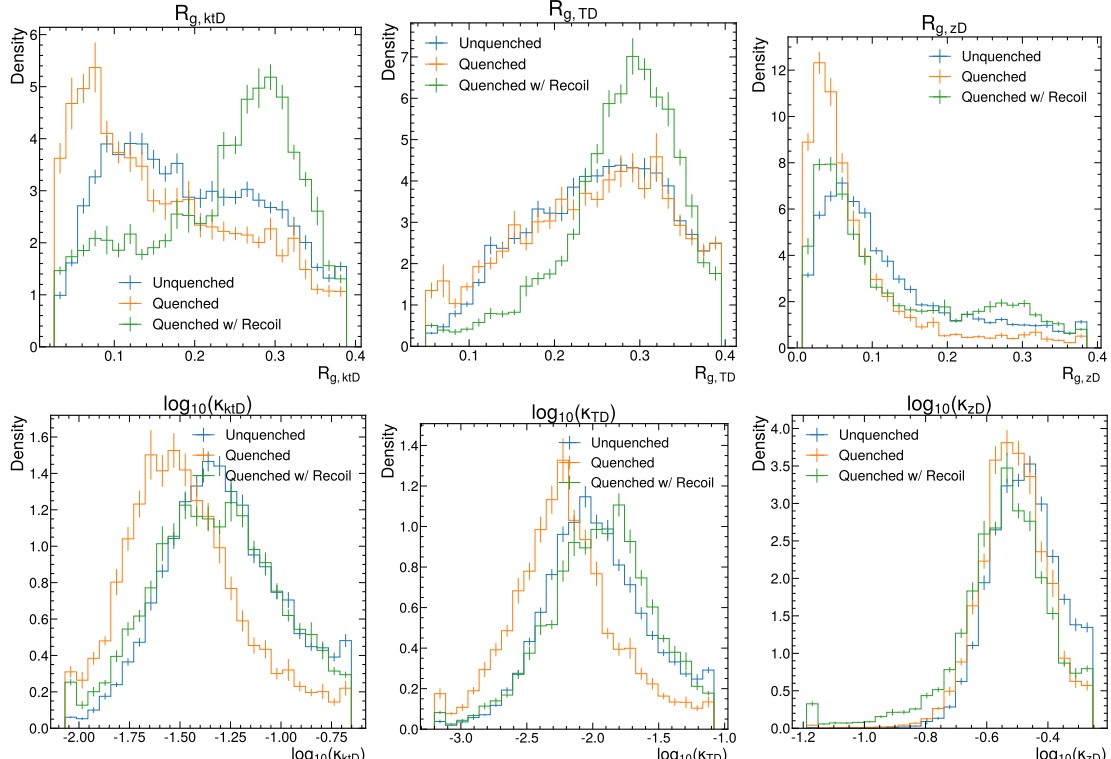

Figure 21: Distributions of example observables for unquenched jets and quenched jets with and without recoil enabled in JEWEL. For the sample with recoil, the thermal background is subtracted using the constituent subtraction procedure from JEWEL [70].

fluctuating underlying event of heavy-ion collisions which cannot be exactly subtracted on an event-by-event basis. This point has been addressed in [73,75] and is beyond the scope of our exploratory study.

In the remainder of this section we explore how our conclusions might change once we produce samples which are closer to the experimental data by including QGP response as modelled by JEWEL. For that, we prepared a Quenched jet sample using JEWEL with recoils, meaning that the medium partons that interact with shower partons are kept in the event record and eventually hadronise together with the shower partons. This introduces additional energy-momentum in the final state, from which the contributions that would have ended up in a jet in the absence of interactions must be subtracted. This is implemented using the JEWEL-specific subtraction algorithm that is based on the event-wise constituent subtraction algorithm [70].

In Fig. 20 we show the obtained distributions for the invariant mass of SoftDropped jets and jet girth, as well as the difference between groomed and ungroomed transverse momentum. For all three of these observables, jet quenching in JEWEL reduces the mean value, while the addition of recoil produces a tail of the distribution that reaches to larger values.

Figure 21 shows a similar comparison for the subjet angles $R_g$ and the $\kappa$ values from three different dynamical grooming strategies. For these observables, the effect of enabling recoil is larger than that of quenching itself and the distributions shift to larger values for the case with quenching and recoils included. This behaviour suggests that these observables are sensitive to soft large-angle radiation and/or background. Importantly, observables where the QGP response results in a significant modification have their distributions moved away from the Unquenched sample.

This first look at the effect of QGP response on some of the most promising jet shape observables shows that a more in depth study of these effects is needed, which is however outside of the scope of the current paper.

# 8   Conclusions and perspective

We carried out three distinct analyses with increasing level of complexity on a set of 31 jet observables computed for both Unquenched and Quenched samples generated with JEWEL+ PYTHIA.

The first, the Principal Component Analysis (PCA) detailed in Section 4, focused on the identification of pairwise linear correlations of observables and their change from the Unquenched to the Quenched sample.

In the second analysis, carried out in Section 5, a Deep Auto-Encoder (AE) which also captures non-linear relations between observables was used to establish the dimensionality of the dataset, that is the number of degrees of freedom needed to encode the information contained in the dataset. Here again, we explored differences between the Unquenched and Quenched samples.

The third analysis, see Section 6, is based on a boosted decision tree (BDT) trained to discriminate the Quenched and Unquenched jet samples using the full set of observables, further exploring the sensitivity of the different observables to jet quenching effects.

Our main findings can be summarized as follows:

- **Subsets of observables are highly mutually correlated**

  In both the Unquenched and Quenched samples, the PCA identified (see Figs. 3 to 5) subsets of mutually correlated observables. In both cases, a large cluster includes observables that capture the transverse substructure of the jet, and jet charges form an independent cluster (they are strongly mutually correlated but uncorrelated with all other observables). Highly correlated observables encode the same information and are thus redundant. This redundancy hints at the possibility of describing the full information content of the set in terms of a reduced number of effective degrees of freedom.

- **The information content of the entire set can be described by a small number of effective degrees of freedom**

  We observed, both for the PCA and the AE, that the number of degrees of freedom needed to account for the full variance of the dataset is rather modest. In the PCA, we found that a small number of principle components is sufficient to capture most the relations between the observables, namely that the first 10 principal components are enough to explain $\sim 90\%$ of the distributions of all observables. Notably, the 6th and 7th principal components capture the distributions of jet rapidity and azimuthal angle which are uncorrelated with both each other and all other observables. This indicates that most physically relevant information is already encoded within the first 5 principal components. In the AE, which also captures non-linear relations between the observables, we found that the relations between the different observables can be captured with a latent space of low dimensionality, i.e. with a small number of nodes in the hidden layer. While the quality of reconstruction, that is to say the ability to predict the distributions for all observables, monotonically increases with increasing dimensionality of the latent space, it saturates at a value close to 100% (fully accurate reconstruction) for dimension 10 being well above 90% for a latent space of dimensionality 5. This indicates that 5 degrees of freedom are sufficient to encode very accurately the full variance of the dataset. The

systematic better quality of reconstruction, see Fig. 13, obtained with the AE in comparison with PCA for the same number of degrees of freedom highlights the importance of non-linear relations among observables.

- **The effective degrees of freedom do not correspond to simple observables**

  The first five principal components, which encode most of the physically relevant information of the dataset, are linear combinations involving most observables (see Fig. 8). As such, an interpretation of a principal component as an observable is not possible. A similar conclusion can be drawn from the AE analysis, where (see Fig. 14) all observables contribute to the expained variance of the dataset.

- **Correlations between observables are mostly resilient to quenching effects**

  While the linear correlation coefficients for some pairs of observables are significantly different for the Quenched and Unquenched sample, the ability to reconstruct the observables in the Quenched sample using the principal components determined in the Unquenched sample (see Fig. 10) indicates remarkable resilience of linear correlations to quenching effects. A notable counterexample is that of the amount of transverse momentum removed by SD, $(\Delta p_\mathrm{T})_{SD}$, which is poorly described in the Quenched sample using the Unquenched PCA. For the AE, which we recall also captures non-linear correlations, the resilience of correlations to quenching effects is enhanced. Here, the AE trained solely on Unquenched events is able to predict very accurately (see Fig. 15) the values for the all the observables in Quenched events.

- **Specific observables and pairs of observables can discriminate between Unchenched and Quenched with similar performance to the complete observable set**

  In Section 6 we compared the ability to discriminate between events in the Unquenched and Quenched samples achieved by BDTs trained on each single observable and on each single pair of observables with the full discrimination power of a BDT trained in all observables (see Fig. 17). Several observables and pairs of observables (a detailed list and discussion can be found in Section 6) were shown to be, by themselves, as discriminant as the full set. These observables, and pairs of observables, are thus optimal candidates for taggers of quenching effects. Importantly, see Fig. 18, observables identified as individually highly discriminant select the same jet population as the all-observable BDT, that is to say they operate the same discrimination as the all-observable BDT.

- **Quenching effects manifest themselves through strong population migration**

  The apparent contradiction between the existence of highly discriminant observables, and pairs of observables, and robustness of pair-wise correlations is resolved (see Fig. 19) by observing that quenching effects strongly modify how the distributions of each observable are populated while maintaining the relation between the observables. Quenching affects mostly the mean or most probable values of observables, not the correlation between pairs.

Overall, our results show that to discriminate quenched and unquenched jets in JEWEL+PYTHIA, single observables or pairs of observables can be chosen that already exhaust the full sensitivity to quenching effects in our studies. The information content redundancy of many of the considered observables provides a guiding principle for experimental measurements. According to our studies, measurement of more than a select few observables has a very limited added value, at least in the context of the jet quenching mechanisms that are implemented

in JEWEL. The ultimate choice among observables and pairs of observables with optimal discriminating power for priority experimental measurement should be dictated by experimental considerations, including their robustness to background subtraction and detected response effects, and achievable precision with recorded collision data.

# Acknowledgments

We are very grateful to Alba Soto Ontoso and Korinna Zapp for providing the code for the dynamical groomer and subtraction scheme, respectively, and giving permission to reshare it alongside our analysis code.

**Funding information** This work is a result of the activities of the Networking Activity 'NA3-Jet-QGP: Quark-Gluon Plasma characterisation with jets' of STRONG-2020 "The strong interaction at the frontier of knowledge: fundamental research and applications" which has received funding from the European Union's Horizon 2020 research and innovation programme under grant agreement No 824093.

JGM further acknowledges the support from Fundação para a Ciência e a Tecnologia (Portugal) under project CERN/FIS-PAR/0032/2021 and the European Research Council (ERC) under the European Union's Horizon 2020 research and innovation programme: Grant agreement No. 835105, YoctoLHC, and gratefully acknowledges the hospitality of the CERN theory group where part of the work took place. The computational work was partially done using the resources made available by RNCA and INCD under project CPCA/A1/401197/2021.

# A  Reproducing this work

The samples used in the analyses presented in this work are available for download in [101]. They can be easily reproduced using the docker images with the necessary software on `gitlab`,[7] which includes instructions on how to run three stages of this work: sample generation, observables computation, and each of the analyses.

All the analyses carried out in this work can also be reproduced by dedicated docker images. Both the codes and the docker images used in this work are available in the repository. We also produced a single script that will sequentially run all the analyses, `run_all.sh`.

# B  Auto-encoder hyperparameter optimisation

To optimise the hyperparameters of the Auto-Encoder we used `optuna`, a framework agnostic hyperparameter optimisation package. For each value of the dimension of the latent space, we optimise for the number of layers, units, and activation function. The hyperparameter space can be seen in Table 3.

We set the number of units and layers for the encoder and decoder to be the same, so that each has the same capacity. We fixed an exponential cyclical learning rate, as this is know to speed up convergence of the training [103], with a fixed cycle over an entire epoch and fixed learning rate decay, `gamma`. We used `BatchNormalization` between each linear layer and the non-linear activation.

The `optuna` search was set to a maximum of 100 trials per dimension of the latent space

---

[7]https://gitlab.com/lip_ml/jet-substructure-observables-ml-analysis.

Table 3: Hyperparameter search space.

| Hyperparameter | Seach Space |
|---|---|
| Optimiser | Adam [102] (Fixed) |
| Encoder Number of Layers | $[1, 10]$ |
| Encoder Number of Units | $[16, 256]$ |
| Decoder Number of Layers | $[1, 10]$ |
| Decoder Number of Units | $[16, 256]$ |
| Activation | {LeakyReLu, PReLu} |
| Weight Initialisation | glorot-normal (Fixed) |
| Learning Rate Scheduler | ExponentialCyclicalLearningRate |
| Minimal Learning Rate | $10^{-8}$ (Fixed) |
| Maximal Learning Rate | $10^{-2}$ (Fixed) |
| Learn Rate Decay (gamma) | 0.9999 (Fixed) |

or a timeout of 60 minutes, whichever came first. The sampler was set to TPESampler with multivariate=True. The best model, for each value of $z_{dim}$, was saved in its best epoch. A median pruner was used to discard unpromising trials, with n_startup_trials=10, n_warmup_steps=10, and interval_steps=5. All these values can be changed using a configuration file, as explained in Appendix A.

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
