# Peer review of "Jet substructure observables for jet quenching in Quark Gluon Plasma: a Machine Learning driven analysis"

_SciPost Physics, doi:SciPost Phys. 16, 015 (2024)_

## Round 1 · Referee Report · Anonymous · 2023-9-28

Strengths
1-Well-organized code and dataset, high reproducibility
2-Clear analysis with good interpretability
3-Clear representation, free of grammar errors
Weaknesses
1-Relatively elementary analysis
2-Little connection to current studies
3-Need to better motivate why more complicated methods are needed
4-Some results not fully included (e.g., accuracy)
Report
The manuscript aims to cover high-level jet sub-structure through machine learning analysis. Section 2 introduces the observables, or features used in the study. Section 3 covers the simulation details used to generate the jet dataset. Sections 4-6 present three different machine learning studies including Principal Component Analysis (PCA), neural network, and boosted tree. These lead to analysis in Quenched/Unquenched discrimination and the impact of quark-gluon plasma (QGP).
Overall, this is a "proof-of-principle" analysis. As the data is high-level, the conclusion is not surprising. While there is some good insight in the draft, the study feels simplistic and serves as a good starting point for further studies. Nonetheless, this is a well-written draft on a new topic. I would be happy to recommend it for publication with a few revisions.
Requested changes
Major:
1-In the introduction, why is a high-level data study relevant? What are the challenges and drawbacks of this consideration?
2-In Section 3, what's the motivation behind splitting the dataset equally?
3-In Section 6, briefly explain ROC and AUC to readers with less background in ML. Why were they chosen instead of just accuracy? What does a score of 0.7 mean?
4-In Sections 6 or 8, a comparison versus traditional methods for Quenched/Unquenched discrimination would be helpful to validate the motivation of the study.
Minor:
- Section 2, first paragraph. What are the assumptions behind the distributions when omitting information?
- Section 2.2, IRC never introduced.
- Section 3, "FastJet Contrib packages..." reformat with proper spacing.
- Section 3, "between in the different dynamical grooming"... remove "in".
- Add commas when necessary, for e.g., "that is, how they change between..."
- "For each, Unquenched and Quenched, sample we compute...," better write "For each sample, Unquenched and Quenched, we compute..."
- Section 4, "The principal component analysis only captures linear relations between observables, which can hide further non-linear relations, ...", using the word "hide" here is incorrect. PCA simply cannot detect non-linear effects.
- Section 5, "This bottleneck layer is usually referred to as the latent space ...", "bottleneck" has a different meaning here, better just used "hidden".
- "For this reason, we developed a hyperparameter optimisation loop using the python package optuna", can just say "we optimised model hyperparameter using the Python package optuna"
- Section 6, "we start by creating a strong baseline by training a BDT using all observables...", rephrase "we create a strong baseline by training ..."
Author: Miguel Crispim Romao on 2023-12-08 [id 4179]
(in reply to Report 1 on 2023-09-28)
Dear referee,
We thank the feedback and the suggested changes, which we address below. Please find attached the latexdiff of the latest version to better identify the changes to the manuscript.
Best regards,
The authors
1-In the introduction, why is a high-level data study relevant? What are the challenges and drawbacks of this consideration?
High-level observables have several advantages. Their use/selection is often motivated by theoretical considerations, due to expected sensitivity to specific effects and calculability. These embody domain-specific knowledge, which also means that the resulting selections on these variables are more readily interpretable than low-level observables (see Section 2 of the paper). Experiments normally report high-level observables, since these can be corrected for experimental effects.
Added text in the manuscript: ‘several of which are directly motivated by theoretical arguments and are commonly reported by the experiments, ‘
2-In Section 3, what's the motivation behind splitting the dataset equally?
We have added to section 3 the following motivation: ‘Since each stage relies heavily on the capacity of the data set to provide a strong statistical representation of the underlying processes, the methodology herein is only as good as its weakest link. To ensure similar statistical strength in each of the steps, samples of equal size are used.’
3-In Section 6, briefly explain ROC and AUC to readers with less background in ML. Why were they chosen instead of just accuracy? What does a score of 0.7 mean?
The section was expanded to include the definition of the ROC, its AUC, and how to interpret the value, as well as a footnote discussing the inadequacy of using accuracy in HEP.
4-In Sections 6 or 8, a comparison versus traditional methods for Quenched/Unquenched discrimination would be helpful to validate the motivation of the study.
Jet quenching is assessed by comparison of distributions of experimentally measured observables in heavy ion collisions and proton-proton collisions (the no-quenching baseline). No procedure exists to establish on a jet-by-jet basis whether that jet has been quenched (modified by its interaction with the quark gluon plasma) or not, or to what degree.
It is worth noting that also in this paper, the discrimination of quenched jets is not a goal per se; we are rather using this question as a way to explore the sensitivity of different observables to jet quenching. Fig 18 in Section 7 compares the selectivity of the full BDT to a more traditional ‘cut-and-count’ approach using 3 of the most sensitive jet shape variables.
Minor:
Section 2, first paragraph. What are the assumptions behind the distributions when omitting information?
No assumptions are made concerning the distribution of each observable. We only consider observables that deliver a single scalar quantity for each jet. That is to say that for each jet within our samples, a number of observables (those listed in sec.2) is computed on a per-jet basis so that each jet is characterized by a vector where each entry is an observable. We do not consider as input the distributions of these observables which are only computable for a set (sample) of jets. The logic behind this approach is that we aim to make statements on a jet-by-jet basis and not on samples of jets
All other Minor were addressed in the manuscript.
Author: Miguel Crispim Romao on 2023-12-08 [id 4178]
(in reply to Report 2 on 2023-10-06)Dear referee,
We thank the feedback and the suggested changes, which we address below. Please find attached the latexdiff of the latest version to better identify the changes to the manuscript.
Best regards,
The authors
Requested changes
1 - The paper is well written and I do not request major changes. However, I'd urge the authors to deliver more discussion to the point #3 I raise in weaknesses.
Added on page 15: ‘This separation of observables in several groups is to some extent in line with expectations: for example the jet charges are explicitly sensitive to the total charge of the jet, and the charge of the parton producing the jet, while the angularities measure the momentum distributions inside jets. The grooming and subjettiness observables are expected to be more sensitive to specific substructure than the angularities. The fact that many observables within each category are grouped together in each principal component suggests that the number of independent degrees of freedom in the data set is much smaller than the number of observables that was studied.’
2 - One minor point is that I have a difficulty of fully capturing the discussion of "correlations between observables are mostly robust to quenching effects". What does this actually mean? The text in this section does not really clarify the point of robustness to quenching. Perhaps authors could expand on this point. Perhaps making the point around the wording "resilient" (in the vicinity of meaning of 'robust') would be better.
The key point to our mind is that the (linear) correlations (coefficients) between pairs or larger groups of observables is similar for quenched and unquenched jets; this is in particular seen in the PCA, where the explained variance of the quenched sample when using the coefficients determined from the unquenched is similar to that using the quenched sample, as mentioned in the text: ‘the ability to reconstruct the observables in the Quenched sample using the principal components determined in the Unquenched sample (see Fig. 10) indicates remarkable robustness/resilience of linear correlations to quenching effects’
We have changed the word robustness to resilience in the manuscript as we agree with the referee that it conveys better the statements we make.
Attachment:
paper-diff.pdf

---

## Round 1 · Referee Report · Anonymous · 2023-10-6

Strengths
1 - The presented work responds well to a clear need for ML study of high-level jet substructure observables and their relative strengths. It contributes to the discussion on efficient (targeted observable use) capturing the nature of interactions of jets within quark-gluon plasma created in high-energy nuclear collisions. The conclusions of the paper may help to steer experimental programs providing a remarkably concise conclusion that measurements "of more than a select few observables has a very limited added value".
2 - The manuscript constitutes a useful report since it presents a rather complete study of variety of observables, both IRC-safe and IRC-unsafe (relevant for completeness as the information on jet-qgp medium interactions and in-medium jet modifications - also referred to as jet quenching - span both regimes) with applications of jet grooming techniques that cover the span of current directions in the field and expose sensitivity to domains accessible and not accessible with perturbative techniques.
3 - The paper employs techniques that attempt to capture linear relation between observables (Principal Component Analysis) but also addresses their non-linear relations (via an autoencoder implementation). It also incorporates a boosted decision tree technique to isolate most disciminating observables for jet quenching and presents an informative "heat map" as a concise summary that can guide experimental research (at the limit of the model dependence and experimental difficulties in performing jet measurements in high energy heavy-ion collisions or the expected precision of the measurements that can be observable-dependent).
4 - The paper is well written with clear goals in mind. It does represent useful conceptual conclusions regarding the correlations of the jet quenching phenomena. The authors employed a thoughtful way of representing the numerical conclusions of the study.
5 - The authors have done a good job enabling the reproducibility of their analysis and the results.
Weaknesses
1 - While the paper focuses on the conceptual development to systematize the information regarding jet quenching based on high-level observables it does so only using a single model (called JEWEL). Some of the conclusions might be model dependent. In particular, this may be of importance for observables that are sensitive to the details of the energy/particle flux excited from the medium by the passage of an energetic jet (coined as QGP response) and - related to it - the quasi-particle (if assumed) implementation of the medium itself (driving the magnitude of quenching effects on a jet-by-jet basis).
2 - As authors note themselves the understanding of experimental applicability of the presented methodology and conclusions are beyond the scope of the paper. However, it remains somewhat important as of what of the presented work will, in the end, be conclusive or applicable at all within the experiments. Having that, not to ponder too hard on this point the work remains useful as it may provide some qualitative guidance for experiments.
3 - The conclusions from the PCA analysis are lacking the reflection on the relation of the variance with respect to the relevant information of jet quenching. Also, there is little discussion on the explainability of the findings. For example, in the case of the conclusion that dynamical grooming observables provide information not captured by angulatiry-type observables the paper lacks a discussion or follow-up on what this information would be. This is likely the consequence of general difficulty of interpretability in the methodology used. We are exposed with an observation of an effect w/o the explanation of it's cause. Nonetheless, the resulting conclusion that very limited number of observables can capture all medium-induced effects is valuable.
Report
In general, the paper satisfies criteria for publication. In particular, I believe the presented work touches on a previously-identified and long-standing research stumbling block - namely, respond to a quite obvious need within the field of rigorous analysis of usefulness of high-level observables for a particular purpose of jet quenching measurements. Moreover, it has a potential to open a new pathway in the existing research direction with enough promise for phenomenological, theoretical, and experimental follow-up work.
I have no need to go into details - I believe the paper fullfiled *all* of the six general acceptance criteria in SciPost Physics.
Requested changes
1 - The paper is well written and I do not request major changes. However, I'd urge the authors to deliver more discussion to the point #3 I raise in weaknesses.
2 - One minor point is that I have a difficulty of fully capturing the discussion of "correlations between observables are mostly robust to quenching effects". What does this actually mean? The text in this section does not really clarify the point of robustness to quenching. Perhaps authors could expand on this point. Perhaps making the point around the wording "resilient" (in the vicinity of meaning of 'robust') would be better.

---

## Round 2 · Referee Report · Anonymous (Referee 1) · 2023-12-14

Report

The submission has addressed all of my concerns. I'm happy to recommend it for publication.

---

## Round 2 · Author Response

Dear editor,

We would like to thank the referees for the positive and constructive feedback. We have followed their suggestions and applied the minor corrections and suggestions proposed to us.

Best regards,
Miguel Crispim Romao, for the authors

---

## Round 2 · List of Changes

Applied all minor corrections and suggestions from the referee.

In particular: changed robust to resilient across the text, extended the discussion on the dataset splits sizes, extended the discussion on the separation of observable types by the PCA, added an explanation of the ROC curve and how to interpret it.

The changes are better seen in the latexdiff provided to the referees in the comments.

---

## Editorial Decision

published